# Buca della Iena and Grotta del Capriolo: New chronological, lithic, and faunal analyses of two late Mousterian sites in Central Italy

Jacopo Gennai[1]*, Tom Higham[2,3], Marco Romboni[4,5], Angelica Fiorillo[6,7], Maddalena Gianni[2,3], Laura van der Sluis[2,3], Damiano Marchi[5,8], Elisabetta Starnini[1]

1 Department of Civilisations and Forms of Knowledge, University of Pisa, Pisa, Italy, 2 Department of Evolutionary Anthropology, University of Vienna, Vienna, Austria, 3 Human Evolution and Archaeological Science (HEAS), Vienna, Austria, 4 Department of Environmental Biology, University of Rome 'La Sapienza', Rome, Italy, 5 Department of Biology, University of Pisa, Pisa, Italy, 6 Department of History, Culture and Society, University of Rome 'Tor Vergata', Rome, Italy, 7 Department of Cultural Heritage, University of Bologna, Bologna, Italy, 8 Centre for the Exploration of the Deep Human Journey, University of the Witwatersrand, Johannesburg, South Africa

* jacopo.gennai@cfs.unipi.it

## Abstract

New radiocarbon, lithic, faunal, and documentary analyses of two sites, Buca della Iena and Grotta del Capriolo, located in Tuscany (Central Italy) and excavated in the late 1960s', are presented. The new analyses significance will be evaluated within the late Neanderthal occupation in the northwestern Italian peninsula and provide insights into their demise. Reassessment of stratigraphical and fieldwork documentation identified areas of stratigraphic reliability, supporting robust interpretations. Radiocarbon dating reveals broadly contemporaneous occupations at both sites between 50–40 ka cal BP, with Buca della Iena showing occupation from approximately 47 to 42.5 ka cal BP. Lithic analyses demonstrate the consistent application of the same *chaîne opératoire* across both sites. Faunal analyses indicate that carnivores, particularly *Crocuta spelaea*, were the dominant accumulating agents in Buca della Iena, while limited preservation at Grotta del Capriolo prevents detailed taxonomic determination. However, hominin presence at both sites is evidenced by cut-marked bones. This study provides new perspectives on the Middle-to-Upper Palaeolithic transition in the northwestern Italian peninsula.

## Introduction

Neanderthal demise is a pivotal topic in palaeoanthropology [1–6]. Despite some possible longer-lasting occurrences [7,8], European-wide dates associated with Neanderthal and archaeological layers bearing their associated material culture, the Mousterian, place their disappearance around 40 ka cal BP [9].

which permits unrestricted use, distribution, and reproduction in any medium, provided the original author and source are credited.

**Data availability statement:** All relevant data are within the manuscript and its Supporting information files.

**Funding:** The research is funded by the Horizon Europe scheme (GA no. 101061427 Acronym: MobiliTy) awarded to Jacopo Gennai and Elisabetta Starnini. Dates of Buca del Tasso and one date of Buca della Iena were funded by the Center Museo di Storia Naturale of the University of Pisa awarded to Damiano Marchi. Funding for the Open Access fee is provided by the Progetto di Eccellenza 2023-2027 'Un senso nel disordine. Praticare la complessità' (CUP I57G23000090006) awarded to the Department of Civilisations and Forms of Knowledge of the University of Pisa by the Italian Ministry of Research (MUR) The funders had no role in study design, data collection and analysis, decision to publish, or preparation of the manuscript.

**Competing interests:** The authors have declared that no competing interests exist.

The Italian Peninsula features a rich record of Neanderthal sites, especially during the last phase of their evolutionary history (early and mid-Marine Isotope Stage (MIS) 3) [10]. Furthermore, the Italian Peninsula is one of the three Southern European peninsulas, which have been argued to provide environmental *refugia* [11,12].

The Italian Peninsula is a key region for investigating the Middle-to-Upper Palaeolithic transition [13]. Stratigraphic evidence from the most complete sequences reveals a technocomplex succession comprising the Mousterian, Uluzzian, and (proto)-Aurignacian.

The late Italian Mousterian is currently associated exclusively with Neanderthal human remains [14]. Lithic evidence indicates that it is predominantly characterised by the Levallois recurrent unidirectional method, complemented by a centripetal modality. However, in the north-westernmost region, this pattern is disrupted by the prominent use of the Discoid method [15–18]. Late Mousterian contexts generally yield radiometric dates consistent with the 45–42 ka BP range [9,19–23].

The Uluzzian is associated with an early *H. sapiens* dispersal and is found solely in the Italian Peninsula and the southern Balkan Peninsula (Peloponnese) [24,25]. Stratigraphically, it caps the Mousterian and represents a clear cultural break [13,17]. The lithic technology of the Uluzzian exhibits a shift towards the predominant use of the bipolar method to obtain blanks for the production of projectile points [26]. It is dated between 43,1–41,4 cal BP (at 68.2% prob.) [19]. Despite the widespread presence of the Uluzzian in the Italian Peninsula, it has not been found in the north-western corner, except for some unstratified contexts in northern Tuscany (Italy) [24]. Also, despite its presence on the central Tyrrhenian coast [27], it is absent from mid-MIS 3 stratigraphical sequences from the same areas [21,28].

The final *H. sapiens* mid-MIS 3 dispersal is associated with the (proto)-Aurignacian technocomplex [29,30], which is always found stratigraphically above the Uluzzian [26,27,31] and is currently dated within the 42–38 ka cal BP span [20,23,28,32–34]. The (proto)-Aurignacian focuses on volumetric bladelet production, distinct from the Mousterian and the Uluzzian [13,17]. Raw material procurement in Riparo Mochi and Riparo Bombrini (Balzi Rossi sites complex) shows imports from Southern France and Central Italy, therefore lending support to the existence of a northern Mediterranean dispersal route [30,35,36].

Neighbouring the Italian Peninsula, the cultural sequence discovered in the Rhône Valley, at Grotte Mandrin, is also noteworthy [37]. The Middle-to-Upper Palaeolithic Transition sequence spans from 54 ka cal BP and includes the Neronian, a material culture distinct from the Mousterian and with affinities to the Initial Upper Palaeolithic. The Neronian has been identified exclusively within the Rhône Valley and has recently been attributed to a short-lived H. sapiens occupation around 54 ka cal BP [37,38]. Subsequently, Neanderthals reoccupied the cave, producing primarily Discoid and Levallois blanks until approximately 42 ka cal BP [37,39]. Finally, the (proto)-Aurignacian caps the stratigraphical sequence, aligning the site with the broader Mediterranean European evidence [37,40]. The origins of the Neronian in the Rhône Valley remain unclear, yet, similarly to the (proto)-Aurignacian [41], it may have involved a circum-Mediterranean dispersal route from the Levant, potentially crossing the northern Italian Peninsula.

Despite the broadly valid chronological sequence (Mousterian – Uluzzian – (proto)Aurignacian), patterns of presence, absence, and replacement are far from linear. As mentioned above, the Uluzzian is absent from the northwestern corner of the Italian Peninsula (Fig 1), the earliest (proto)-Aurignacian is not ubiquitous either, and some sites hint at the survival of Mousterian (Neanderthal) pockets up to ~40–41 ka cal BP [21–23,35,42,43].

Northwestern Tuscany (Italy) could be one of these pockets. Here, there are at least five stratified sites, which are dated and attributed typologically to the final Mousterian [21,46–48]. As most of these archaeological assemblages were excavated during the first half of the 20th century, low finds densities and the limited extent of the sites have hampered a precise understanding, resulting in the sites slowly being forgotten or data being discarded due to the many uncertainties [21,48–52]. As no other MIS 3 material culture was discovered at the sites, the earliest frequentations after the Mousterian, if occurring, are the Epigravettian, the Neolithic or the Metal Ages [51,53–55]. Buca della Iena and Grotta

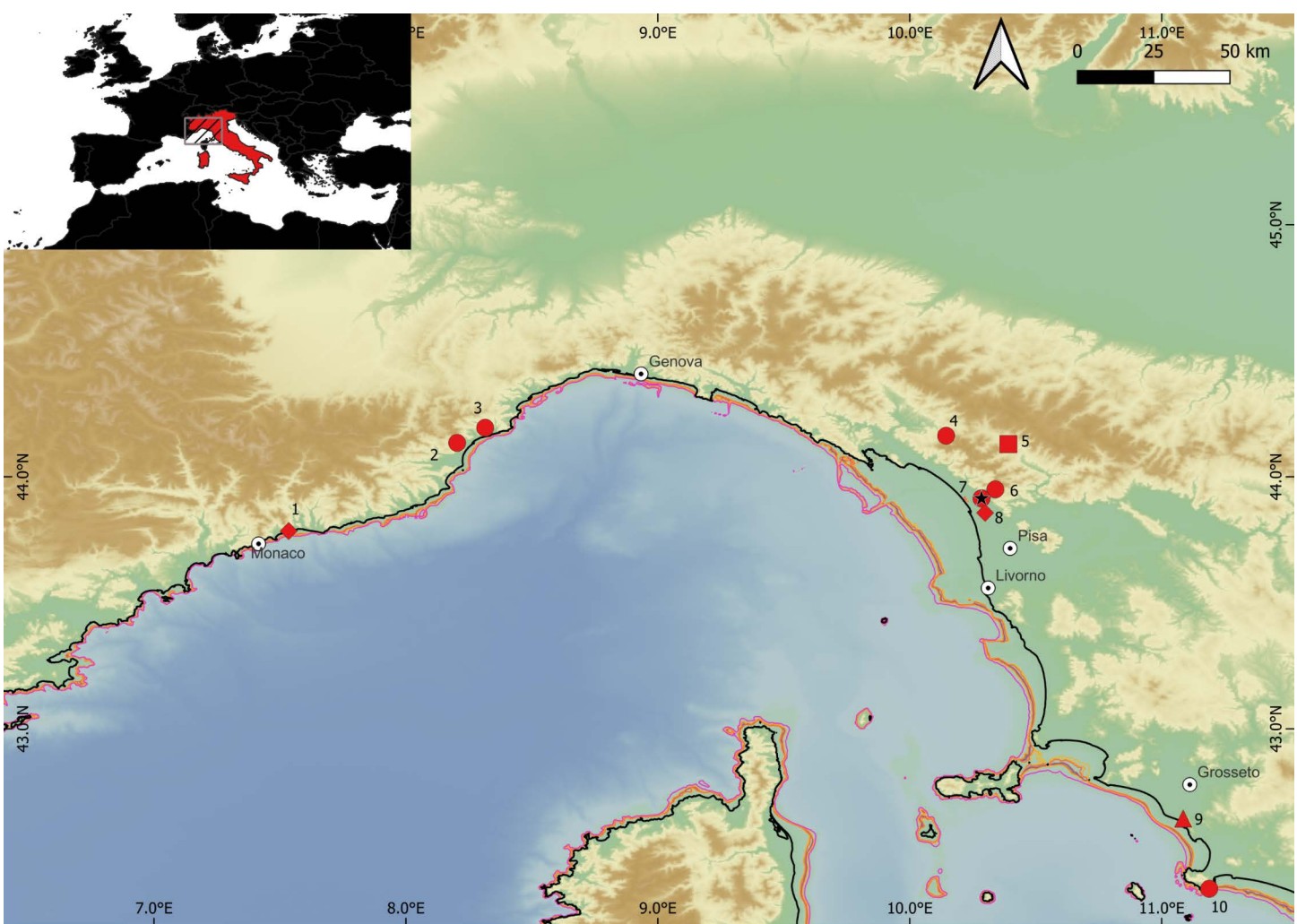

**Fig 1. The northwestern Italian Peninsula area and the mentioned sites.** Symbols refer to the MIS 3 chronocultural phases present on-site: Circle, only Mousterian, Lozenge, Mousterian and Aurignacian, Triangle, Mousterian, Uluzzian, and Aurignacian. 1: Balzi Rossi, 2: Santa Lucia Superiore, 3: Arma delle Manie, 4: Tecchia d'Equi, 5: Pontecosi, 6: Grotta all'Onda e Buca del Tasso, 7: Buca della Iena e Grotta del Capriolo, 8: Massaciuccoli, 9: Grotta La Fabbrica, 10: Grotta dei Santi. MIS 3 eustatic sea level lines from 55 to 35 ka cal BP [44] are reported on the modern bathymetry. Black solid line is the modern coastline. Map created with QGIS 3.36 Maidenhead, elevation data elaborated from GEBCO_grid 2023 [45].

del Capriolo (Fig 2) could shed light on the last Neanderthal occupation of the area as they are neighbouring sites and the Buca della Iena archaeological sequence lies upon a flowstone which has been dated to <41 - <51 ka BP [56,57]. Both sites are better understood as karst openings with archaeological infilling or collapsed rock shelters with niches. We revisited the two sites to reanalyse the surviving excavation documentation and lithic assemblages, conduct new faunal taxonomic and taphonomic analyses, and obtain radiocarbon dates from newly sampled material. This paper presents the results of these revisions, with a particular focus on the new radiocarbon dates and their implications.

## Materials and methods

The new investigations involved a comprehensive understanding of the excavation history and excavation fieldnotes, analysis of the lithic artefacts, analysis of fauna bones, sampling and dating of fauna bones. The Soprintendenza Archeologia, Belle Arti e Paesaggio Lucca e Massa Carrara issued all necessary permits for the described study.

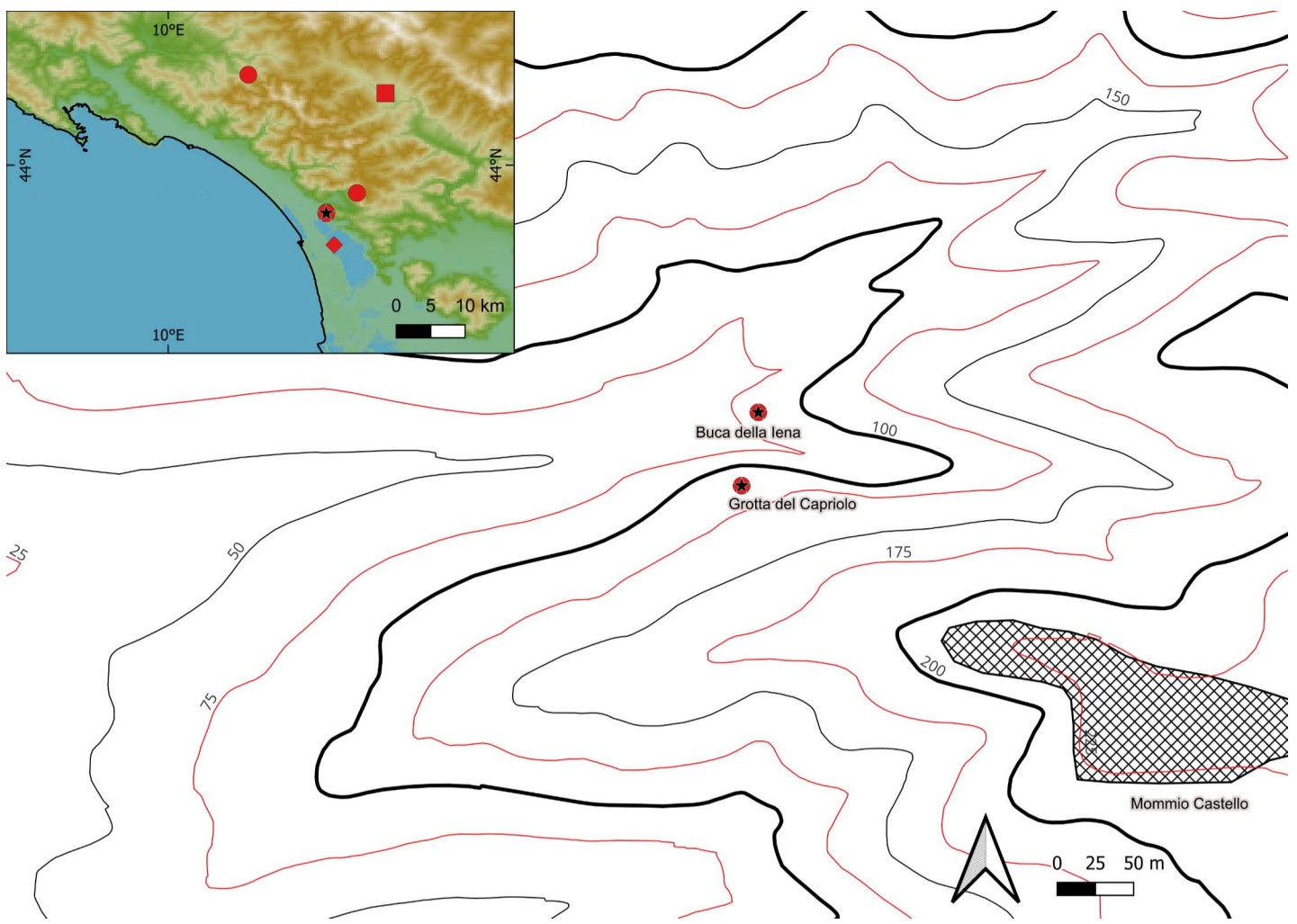

**Fig 2. Northwestern Tuscany and the exact location of Buca della Iena and Grotta del Capriolo (red dot and black star symbol).** Map created with QGIS 3.36 Maidenhead, the GEBCO_Grid 2023 [45].

## Sites

**Buca della Iena** (43.9142645 N, 10.2850590 E; 85 m a.s.l.) was discovered and completely excavated in 1966, which led to the publication of the results in 1971 [56,58]. Additional excavations occurred in 1972 and 1973, dismantling completely the surviving northern profile. Currently, the excavation has been backfilled and only the karst opening is visible.

The excavation proceeded in arbitrary spits (50–10 cm thick) and different trenches (A – E). According to the sedimentological content, the excavators reconstructed five main lithological layers:

- Humic level (h)

- Silty, sandy brown-reddish loose sediment with gravel (Layer A)

- Greyish silty, sandy sediment with gravel: the gravel increases towards the bottom and the sediment is progressively more compact (Layer B). Pitti and Tozzi [56] refined the knowledge of the main deposit (layer B), allocating subdivisions according to the gravel content and progressive hardening of the sediment (from bottom to top B3, B2, B1).

- Flowstone

- Yellowish loamy sediment with abundant animal bones (layer D).

Through the analysis of the original fieldwork documentation, it is possible to improve the understanding of the stratigraphical sequence, and most importantly, the recovery context of the archaeological findings [59,60]. In Buca della Iena only findings from sectors F, C, and D can be traced back to a consistent stratigraphical position. As reported in the fieldwork documentation Sectors E, A, and B are disturbed or suffered slope and erosional process. Furthermore, most of the lithostratigraphic layer A consists of reworked sediment. Overall, we have more confidence in the correlations for trenches C, D, and F, which were adjacent to the final depicted profile (Fig 3). Hence, throughout the analysis, we refer to levels that are formed by spits with the same number in trenches C, D and F (i.e., level 7 consists of spit 7 in trenches C, D, and F; S8 File).

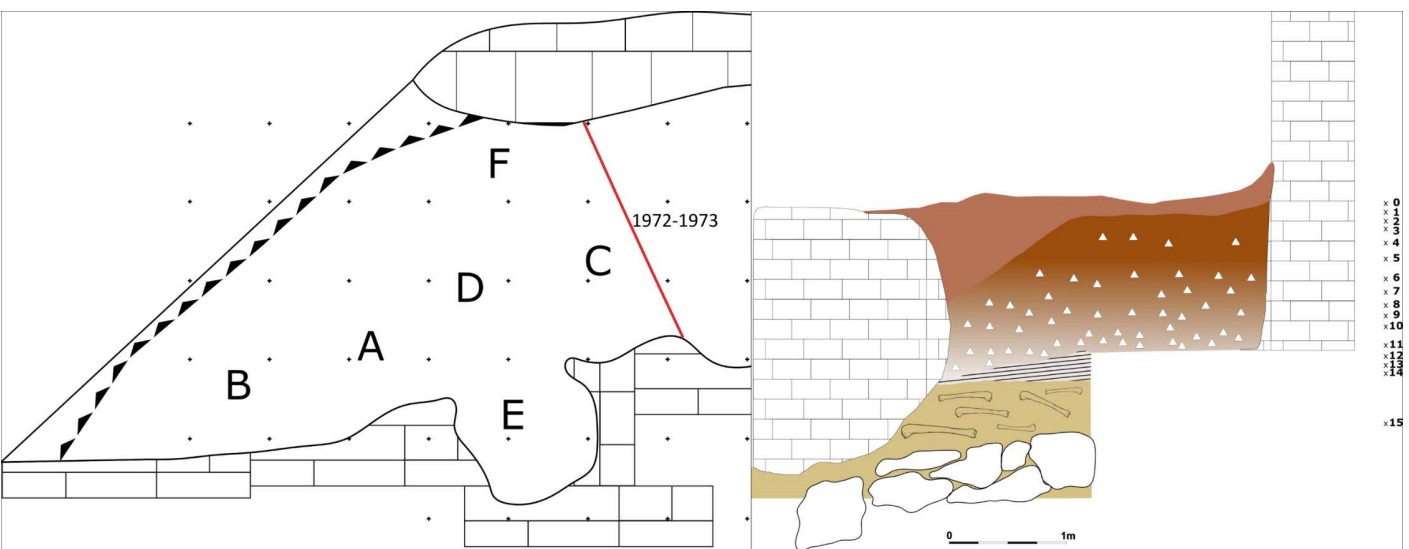

**Fig 3. Buca della Iena reconstructed trenches and the northern profile sketch.** The grid is 1x1m. The x and number indicate the approximate positions of spits. Redrawn and modified by Jacopo Gennai [60].

**Grotta del Capriolo** (43.9136514 N, 10.2847033 E, 90 m a.s.l.) is on the opposite canyon slope facing Buca della Iena (Fig 2). The site was discovered in 1968 and the deposit was completely excavated in 1968 and 1970 [58]. The whole sequence consists of 4 main lithological layers:

• Superficial humic level, 5–20 cm (h)

• Brown-reddish silt with little gravel and devoid of any archaeological material (A)

• Light brown silt with progressively increasing gravel bearing archaeological Mousterian artefacts and Pleistocene fauna (B, subdivisions from bottom to top B3, B2, B1)

• Yellow silt covering clast-supported sediment and big boulders interpreted as the bottom of the sequence (C).

The excavation proceeded in arbitrary spits (40–10 cm thick), different trenches (A – E), and different sectors within trenches (1–5) (Fig 4). A correlation between the spits at the same deposit height and the creation of new levels spanning the whole site has been attempted (S9 File).

Nevertheless, only sectors A2 and A3 are likely to be affected by the least amount of post-depositional processes as the rest of the sediment shows sloping (Fig 5).

## Lithic analysis

The lithic assemblages are stored at the Museo Archeologico e dell'Uomo "A.C. Blanc" of Viareggio (LU). Analysis Few artefacts are displayed in showcases and most of the assemblage is in storage: all the artefacts have been analysed regardless of their attribution to unreliable stratigraphical areas. The lithic assemblages have been investigated through the *chaîne opératoire* approach [61]. To illustrate and provide quantifiable and reproducible observations, standard morphological and metrical attributes have been recorded for each artefact [62,63]. Artefacts have been subdivided into raw material units (RMUs) according to their macroscopic characteristics: mostly according to colour, grain size and texture [64].

Additionally, we calculated lithic volumetric densities for each level within the stratigraphically reliable areas to provide an approximate estimate of human presence within the sequences. Following standard calculations, lithic volumetric density is expressed as the total number of lithic artefacts divided by the cubic metres of each stratigraphical unit (level) [65,66]. To estimate the cubic metres of each level, we applied the standard volumetric formula:

$$Volume = Length \times Width \times Height$$

Where Length and Width correspond to the excavated sectors, and Height represents the average thickness of the stratigraphical level.

## Faunal analysis

**Sampling.** The study involved 3931 middle- and large-sized mammal remains from Buca della Iena (=3710) and Grotta del Capriolo (=273), stored at the Museo Archeologico e dell'Uomo "A.C. Blanc" of Viareggio (LU). Mary Stiner analysed Buca della Iena faunal assemblage, nevertheless she reported stratigraphical units or levels that are difficult to interpret and understand with the current state-of-the-art [67,68]. Hence, we are unable to compare her results with ours. The case of Buca della Iena as a hyena den has been considered as an example in several other studies [67,69–72]. Due to the large sample numbers, only specimens coming from stratigraphically reliable excavation areas have been analysed. Hence, Buca della Iena's sectors C, D, and F and Grotta del Capriolo's sectors A2 and A3.

The faunal remains have been analysed for Taxonomic identification [73,74], the determination of the number of identified specimens (NISP; [75] and the minimum number of individuals (MNI; [76]. The estimated age of death is provided according to available criteria [77–83]. Vertebral and rib fragments, which could not be taxonomically attributed,

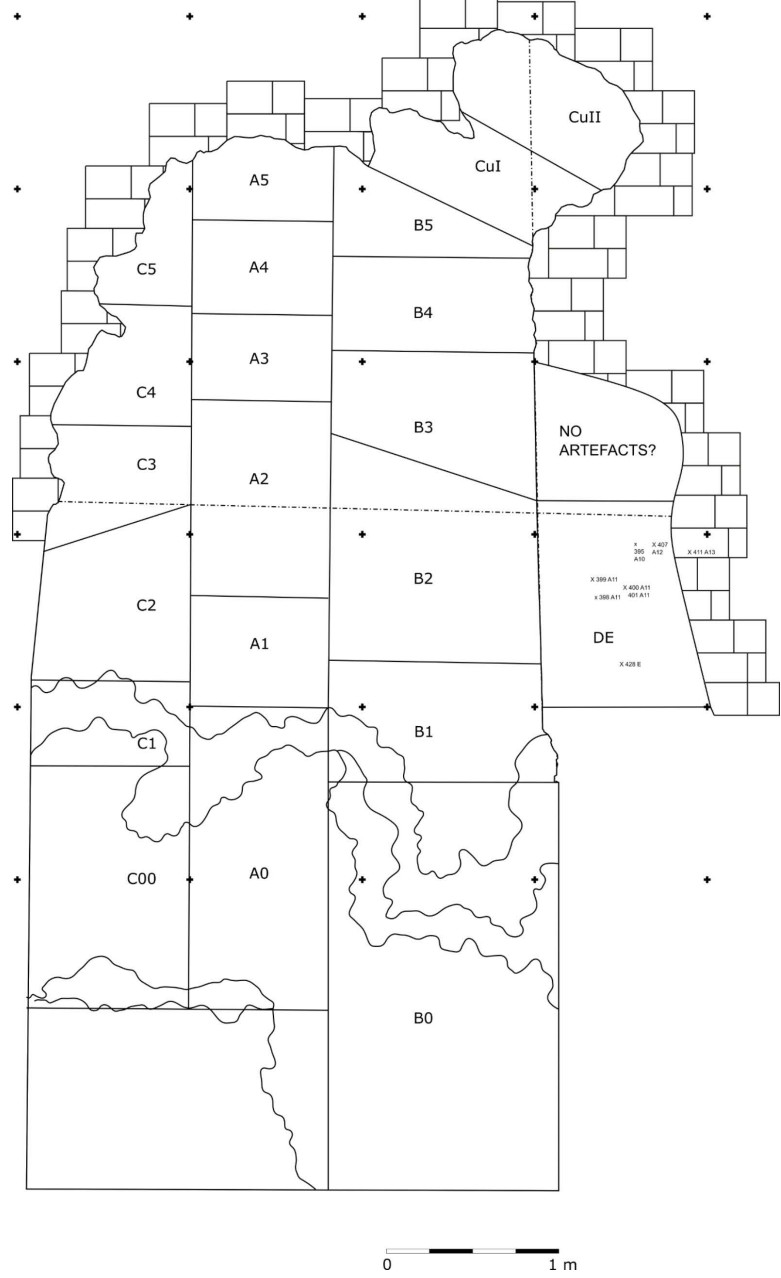

**Fig 4. Grotta del Capriolo excavation trenches.** A 1 × 1 m grid is superimposed. Sourced from [60].

were sorted by size into three categories: small (i.e., *Erinaceus europaeus, Lepus* sp., *Marmota marmota, Mustela* sp.), medium (i.e., *Canis lupus, Meles meles, Panthera pardus, Crocuta spelaea, Sus scrofa, Capreolus capreolus*), and large (i.e., *Ursus spelaeus, Ursus arctos, Palaeoloxodon antiquus, Rhinocerotidae, Equus ferus, Megaloceros giganteus, Cervus elaphus, Bos primigenius*).

The taphonomic study was carried out on all the identified and unidentified (≥1 cm long) remains using a Leica S9I Stereomicroscope and a 10x magnifier to identify both biostratinomic and diagenetic alterations [84]. Regarding

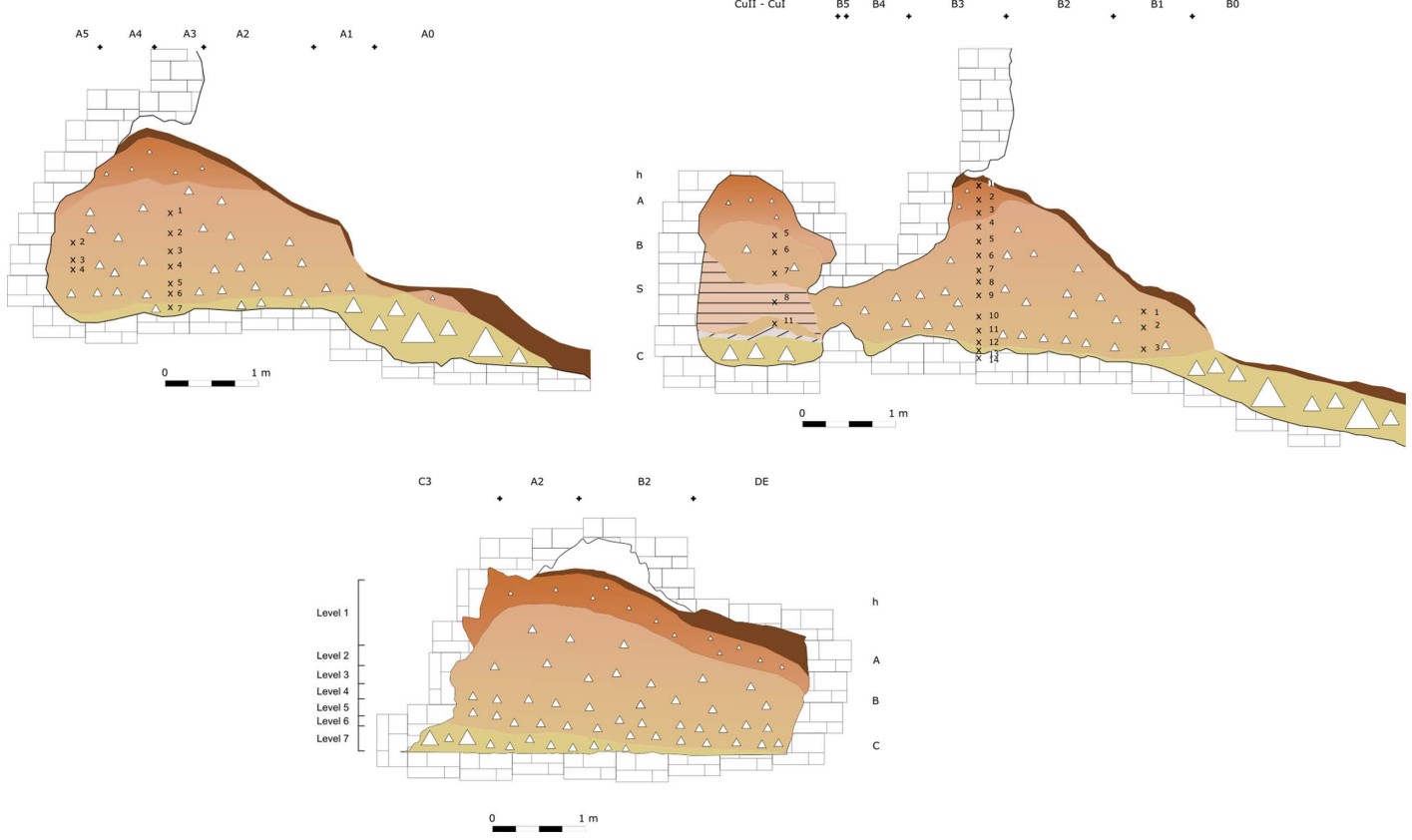

**Fig 5. Grotta del Capriolo profile sections sketches.** Top left: Trench A, Top right: Trench B, Bottom: Transversal section across the three sections. The x and number indicate the approximate positions of spits. On the left side of the transversal section the newly reconstructed levels' positions are shown. Redrawn and modified by Jacopo Gennai [60].

anthropogenic modification, butchery marks, thermoalterations and colour degrees were recorded [85–88]. Carnivore modifications were identified as deep punctures, pits, and scores with U-shaped striations [84,89–93]. Digestion modifications, gnawing by rodents and small carnivores, and ichnotraces by saprophagous insects were also examined [94–96]. Biological and physicochemical alterations, such as root etching and formation of mineral coatings (mainly mineral manganese) were also recorded [94–96].

### Radiocarbon dating

**Sample selection.** Nine faunal samples from Buca della Iena and twelve samples from Grotta del Capriolo were selected, to cover the whole extent of the archaeological sequences. Sampling favoured anthropogenically modified specimens, but unmodified and indeterminable ones were also selected given the rarity of the former. As all the Buca della Iena samples passed the nitrogen content pre-screening test [97], we selected the six samples with the highest %N content (Table 1). Only four samples from Grotta del Capriolo produced a high enough nitrogen content to proceed with radiocarbon dating (Table 1).

### Chemical pretreatment, combustion, graphitisation and AMS measurement

Bone and tooth samples were chemically pretreated and prepared at the Higham Laboratory, University of Vienna. Samples were mechanically cleaned using a tungsten carbide drill, as well as superficially cleaned with a shotblaster, after

**Table 1. Samples of bone tested for collagen content prior to radiocarbon dating.**

| Site | R number | Original ID | Material | Species | Context | Conserved | Modification | Sampled amount (mg) | Amt% N | Amt% C | C/N | Comment |
|---|---|---|---|---|---|---|---|---|---|---|---|---|
| Buca della Iena | R00252 | BdI-1 | Bone | *P. pardus* | Sector D, spit 9 | N | None | 0.84 | 0.56 | 3.33 | 6.9 | %N<0.7 but could be dated without ultrafiltration |
| Buca della Iena | R00253 | BdI-2 | Bone | *M. giganteus* | Sector D, spit 5 | N | None | 1.05 | 1.18 | 7.13 | 7.1 | Suitable for dating |
| Buca della Iena | R00254 | BdI-3 | Bone | *C. crocuta* | Sector D, spit 5 | N | Cutmarks | 0.81 | 2.06 | 7.61 | 4.3 | Suitable for dating |
| Buca della Iena | R00255 | BdI-4 | Bone | ND | Sector E, spit 6 | N | Cutmarks | 0.81 | 0.73 | 4.77 | 7.6 | Suitable for dating |
| Buca della Iena | R00256 | BdI-5 | Bone | *C. elaphus* | Sector F, spit 15 | N | Cutmarks and gnawing | 0.93 | 0.85 | 4.29 | 5.9 | Suitable for dating |
| Buca della Iena | R00257 | BdI-6 | Bone | *C. lupus* | Sector F, spit 14 | N | None | 0.9 | 1.01 | 4.83 | 5.6 | Suitable for dating |
| Buca della Iena | R00258 | BdI-7 | Tooth | *E. ferus* | Sector D, spit 12 | N | None | 1.08 | 0.42 | 3.81 | 10.7 | %N<0.7 but could be dated without ultrafiltration |
| Buca della Iena | R00259 | BdI-8 | Tooth | *E. ferus* | Sector D, spit 9 | N | None | 0.82 | 0.80 | 4.05 | 5.9 | Suitable for dating |
| Buca della Iena | R00260 | BdI-9 | Tooth | *E. ferus* | Sector F, spit 14 | N | None | 0.91 | 0.30 | 3.11 | 12.0 | %N<0.7 but could be dated without ultrafiltration |
| Grotta del Capriolo | R00261 | Gca-1 | Tooth | *E. ferus* | Sector B3, spit 7 | N | None | 1.12 | 0.05 | 3.40 | 72.4 | As %N is significantly lower than 0.7 dating is not recommended |
| Grotta del Capriolo | R00262 | Gca-2 | Tooth | *E. ferus* | Sector B4, spit 7 | N | None | 1.17 | 0.25 | 3.22 | 14.8 | %N<0.7 but could be dated without ultrafiltration |
| Grotta del Capriolo | R00263 | Gca-3 | Bone | herbivore | Sector B2, spit 8 | N | None | 1.16 | 0.03 | 2.47 | 100.9 | As %N is significantly lower than 0.7 dating is not recommended |
| Grotta del Capriolo | R00264 | Gca-4 | Bone | ND | Sector B3, spit 9 | N | None | 1.07 | 0.32 | 1.88 | 6.8 | %N<0.7 but could be dated without ultrafiltration |
| Grotta del Capriolo | R00265 | Gca-5 | Bone | ND | Sector A2, spit 4 | N | Cutmarks | 0.88 | 0.19 | 2.95 | 18.6 | As %N is significantly lower than 0.7 dating is not recommended |
| Grotta del Capriolo | R00266 | Gca-6 | Bone | ND | Sector A2, spit 4 | N | Cutmarks | 0.87 | 0.53 | 3.89 | 8.6 | %N<0.7 but could be dated without ultrafiltration |
| Grotta del Capriolo | R00267 | Gca-7 | Bone | ND | Sector A3, spit 4 | N | Cutmarks | 1.01 | 0.02 | 3.56 | 185.5 | As %N is significantly lower than 0.7 dating is not recommended |
| Grotta del Capriolo | R00268 | Gca-8 | Bone | ND | Sector C2, spit 6 | N | None | 0.84 | 0.03 | 1.90 | 74.8 | As %N is significantly lower than 0.7 dating is not recommended |
| Grotta del Capriolo | R00269 | Gca-9 | Bone | ND | Sector A2, spit 2 | N | None | 0.93 | 0.01 | 2.23 | 251.3 | As %N is significantly lower than 0.7 dating is not recommended |
| Grotta del Capriolo | R00270 | Gca-10 | Bone | ND | Sector B3, spit 9 | N | None | 0.87 | 0.09 | 3.30 | 40.6 | As %N is significantly lower than 0.7 dating is not recommended |
| Grotta del Capriolo | R00271 | Gca-11 | Bone | ND | Sector B3, spit 11 | N | None | 0.92 | 0.31 | 3.63 | 13.5 | %N<0.7 but could be dated without ultrafiltration |
| Grotta del Capriolo | R00272 | Gca-12 | Bone | ND | Sector A3, spit 2 | N | None | 0.83 | 0.03 | 2.98 | 133.1 | As %N is significantly lower than 0.7 dating is not recommended |

Samples with a %N value >~0.7% have a >70% chance of yielding an acceptable yield of collagen for AMS dating [97] C:N atomic ratios on whole bone <17.0 give a 71% success rate in terms of collagen extraction [97].

which samples were crushed into smaller pieces. Laboratory surfaces and equipment were cleaned between sampling different specimens to avoid contamination.

Sample pretreatment for bones and teeth follows collagen extraction (ABA treatment) and ultrafiltration protocols [98,99]. Extracted collagen samples were weighed out in pre-cleaned tin capsules and either combusted and graphitised using the AGE3 system, followed by AMS measurement at the VERA (Vienna Environmental Research Accelerator) AMS facility, University of Vienna [100,101], or at the Keck AMS facility, University of California at Irvine [102,103]. Isotopic fractionation has been corrected for using the $\delta^{13}$C values measured on the AMS and radiocarbon dates are reported in radiocarbon years BP (Before Present - AD 1950) using the half-life of 5568 years.

The extracted collagen was also subjected to stable isotope analysis using an EA-IRMS (Thermo Scientific EA-Isolink with a Flash 2000 coupled to a Delta V Advantage isotope ratio mass spectrometer) in the Silver Laboratory (Large-Instrument Facility for Advanced Isotope Research) at the Center of Microbiology and Environmental Systems Science of the University of Vienna. Stable isotope values of all samples are measured relative to the laboratory standard alanine. $\delta^{13}$C and $\delta^{15}$N values are reported relative to the VPDB and AIR standard, respectively, and are measured to a precision of ±0.3‰.

## Results

### Lithic assemblage

**Buca della Iena.** The Buca della Iena assemblage consists of 157 artefacts. 84 artefacts belong to stratigraphically safe areas (Figs 6 and 7), while 73 are from the reworked sediment (Fig 8 and Table 2). No lithic artefact has been retrieved from the lowermost level 15 (S10 File).

**Taphonomy.** The artefacts do not show extensive patina. Edge damage is more diffused, nevertheless ridges are sharp. Complete artefacts are the most frequent.

**Taxonomy.** Despite the stratigraphic reliability, the assemblage is rather homogeneous in its technotypological characteristics. All phases of reduction are represented (Table 2 and Figs 6–8).

Flakes are the most frequent product. Cortex is found on slightly less than a third of the assemblage. The categories showing the most cortical surfaces besides cortical flakes are the core-edge and overshot flakes. Reduction methods are either Levallois or Discoid. Levallois is prevalent, also with artefacts belonging to convexities management stages, such as core-edge and overshot flakes. Most of the artefacts show a unidirectional mode of flaking. Retouched artefacts amount to roughly a quarter of the blanks and are mostly present within the Levallois blanks. Only levels 10, 6 and 5 have cores. Blades are present only in levels 10, 9, and 8 (Fig 9). The frequency of types does not show meaningful variation throughout the sequence. In most of the levels, there is a co-presence of simple flakes, Levallois recurrent flakes and Levallois reduction core-edge and overshot flakes (Fig 10). The artefact density peaks in level 14 (>15 artefacts per cubic metre) and is lowest in level 13. Other high peaks are represented by level 10 and level 5, while level 7 and level 11 show low peaks (Fig 11). The highest frequency of retouched artefacts is found respectively in levels 12, 5, and 11 where the frequency of retouched artefacts reaches over a third of the relative totals.

### Grotta del Capriolo

The Grotta del Capriolo assemblage consists of 539 artefacts, of which 99 belong to stratigraphically reliable areas (Figs 12 and 13), while 440 are from the reworked sediment, surface scatters or areas affected by sediment sloping (Fig 14 and Table 3).

**Taphonomy.** The artefacts do not show extensive patina, although some have calcareous encrustations. Edge damage is more diffused, nevertheless, ridges are mostly sharp. Complete artefacts are the most frequent. Two fragments found in sectors B2 and B4 spit 10 conjoin, forming a Levallois recurrent flake (Fig 14). Another Levallois flake found in sector A2 spit 6 conjoins with a fragment found at the back of the cave in reworked sediment.

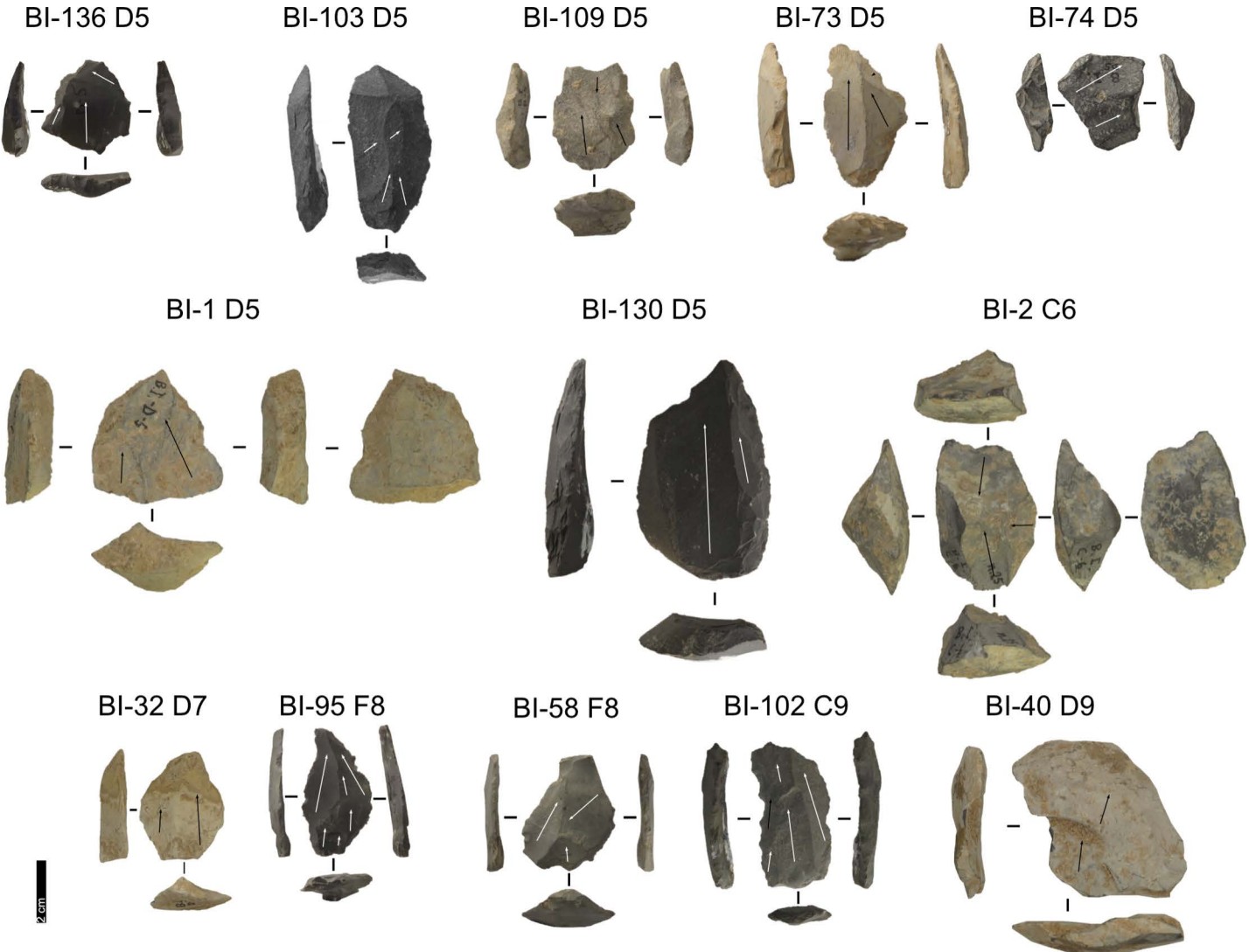

**Fig 6. Buca della lena lithic artefacts from levels 5 to 9.** IDs above each artefact correspond to IDs reported in S2 and S3 Files. Core: BI-1, BI-2; Core-Edge Flake: BI-109, BI-130 (Levallois); Flake: BI-32; Levallois Blade: BI-102; Levallois Recurrent Flake: BI-103, BI-136, BI-58, BI-73, BI-95; Overshot Flake: BI-40; Pseudo-Levallois point: BI-74. Pictures Jacopo Gennai.

**Taxonomy.** Despite the stratigraphic reliability, the assemblage is rather homogeneous in its technotypological characteristics. All the phases of the reduction are represented (Table 3 and Figs 12–14). Flakes are the most represented blanks. Cortex is found on slightly more than a third of the assemblage. The categories showing the most cortical surfaces besides cortical flakes are the core-edge and overshot flakes. Reduction methods are either Levallois and Discoid, Levallois is prevalent with core-edge and overshot artefacts. Additionally, the volumetric method is present with cores and few blanks. Most of the artefacts show a unidirectional mode of flaking. Only level 2 has cores, and blades are variably present throughout levels 1–5, with the highest frequency in level 5 (Fig 15). The frequency of types does not show meaningful variation throughout the sequence, in most of the levels there is a co-presence of simple flakes, Levallois recurrent flakes and their core-edge and overshot products (Fig 16). Retouched artefacts are rare, the highest frequency

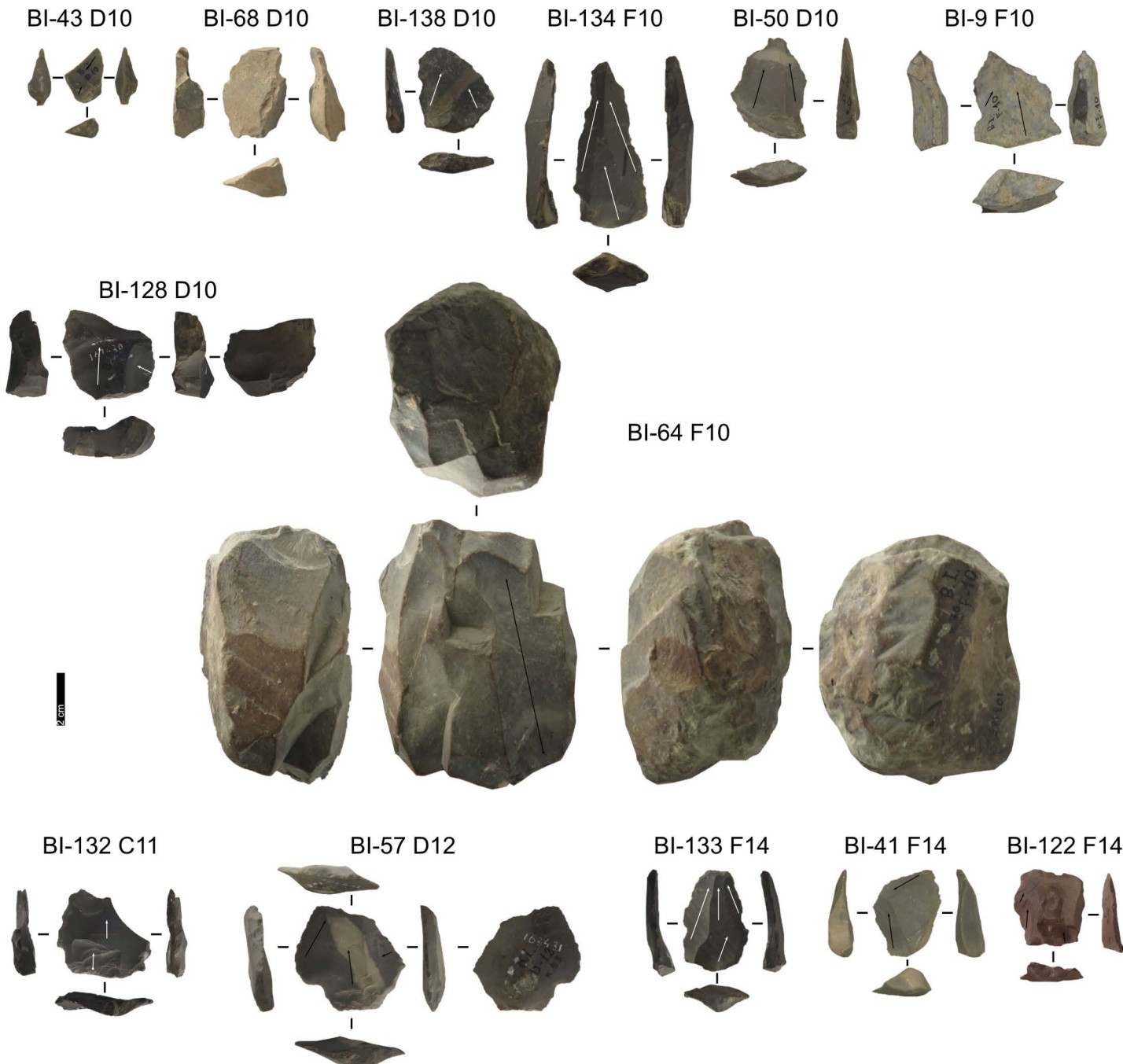

**Fig 7. Buca della lena lithic artefacts from levels 10 to 14.** IDs above each artefact correspond to IDs reported in S2 and S3 Files. Core: BI-64, BI-128; Core-Edge Flake: BI-41 (Levallois), BI-68 (Discoid); Cortical Flake: BI-122; Discoid Flake: BI-9; Levallois Blade: BI-134; Levallois Recurrent Flake: BI-132, BI-133, BI-138, BI-50, BI-57; Pseudo-Levallois point: BI-43 (Discoid). Pictures Jacopo Gennai.

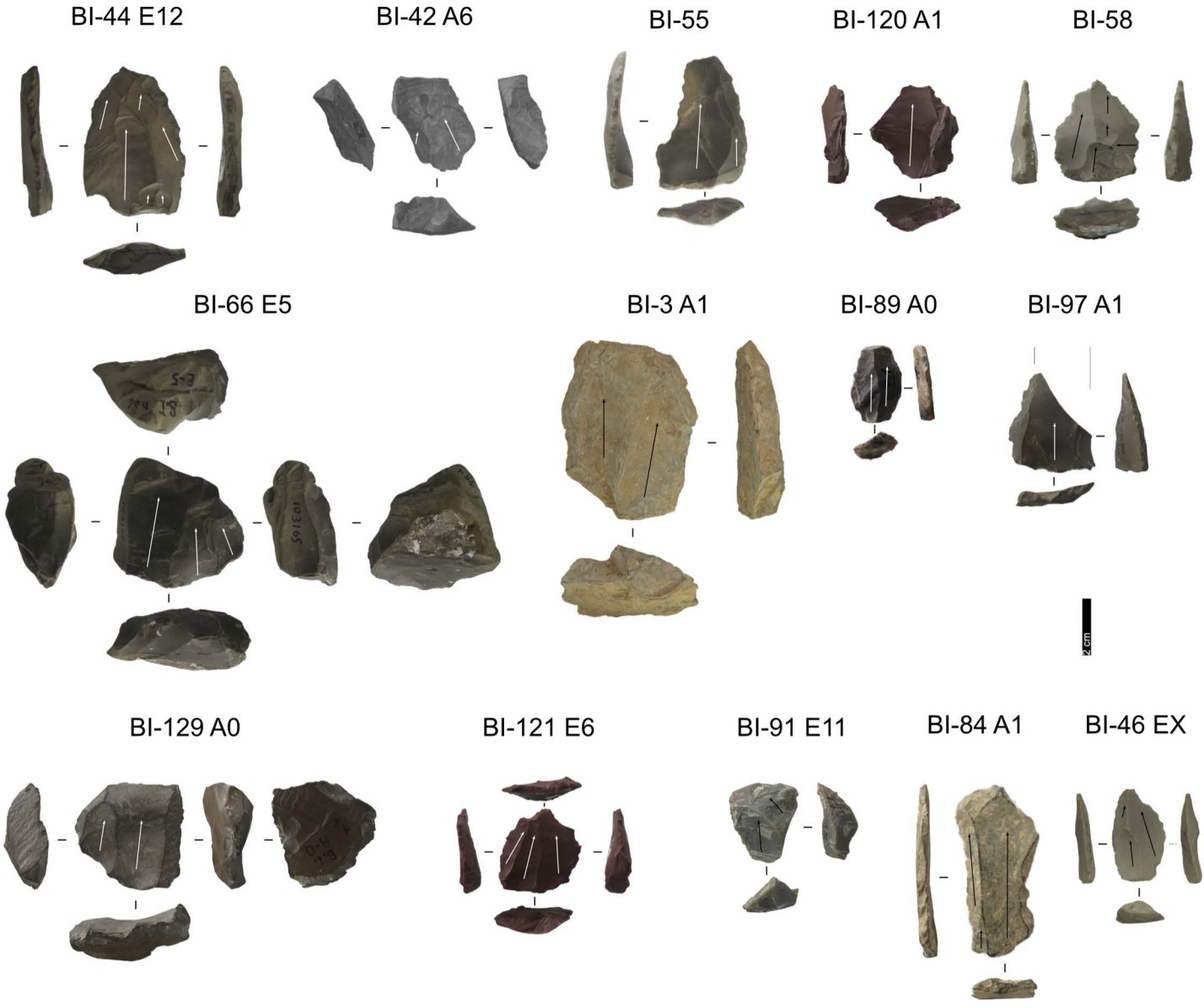

**Fig 8. Buca della Iena lithic artefacts from stratigraphically unreliable context.** Core: BI-66, BI-129; Core-Edge Flake: BI-120; BI-42 (Discoid); Cortical Flake: BI-3; Levallois Blade: BI-84; Levallois Recurrent Flake: BI-121, BI-44, BI-46, BI-55, BI-89, BI-97; Pseudo-Levallois point: BI-91. Pictures Jacopo Gennai.

of retouched artefacts is amongst the elongated artefacts reaching a fifth of the items, mostly within Levallois blades and volumetric blades. The artefact density reaches the highest peak in Level 2 (>90 artefacts per cubic metre) and reaches the lowermost in Level 7. Other high peaks are represented by level 4, while levels 1, 3, 5 and 6 show lower densities (Fig 17). The frequency of retouched artefacts is lower in stratigraphical reliable sectors than within the whole assemblage. Retouched artefacts occur only in levels 1, 2, and 5, with the highest frequency in the latter (11%).

**Raw materials.** Both Buca della Iena and Grotta del Capriolo assemblages show a similar wide array of microcrystalline siliceous material. The preliminary raw material analyses show the presence of mostly calcilutites,

**Table 2. Buca della Iena lithic artefacts categories and counts.**

| Buca della Iena | Stratigraphically unreliable | Stratigraphically reliable | Total |
|---|---|---|---|
| **Blade** | | | |
| *Volumetric Blade* | 1 | | 1 |
| *Cortical Flake* | 1 | | 1 |
| Laminar Asymmetrical | 2 | | 2 |
| *Levallois Blade* | 4 | 3 | 7 |
| **Total Blades** | **8** | **3** | **11** |
| **Total Bladelets** | **1** | | **1** |
| **Core** | | | |
| *Secant unifacial* | | 1 | 1 |
| *Prismatic* | | 1 | 1 |
| *Pyramidal* | | 1 | 1 |
| *Secant core* | 3 | | 3 |
| *Surface core* | | 2 | 2 |
| **Total Cores** | **3** | **5** | **8** |
| **Flake** | | | |
| *Cortical Flake* | 3 | 5 | 8 |
| *Predetermining Flake* | 2 | 3 | 5 |
| *Pseudo-Levallois point* | 2 | 2 | 4 |
| Discoid concept | | 1 | 1 |
| Not assigned | 2 | 1 | 3 |
| *Core-Edge Flake* | 4 | 9 | 13 |
| Discoid concept | 1 | 1 | 2 |
| Levallois concept | 1 | 4 | 5 |
| Not assigned | 2 | 4 | 6 |
| *Overshot Flake* | 2 | 5 | 7 |
| Levallois concept | 1 | | 1 |
| Not assigned | 1 | 5 | 6 |
| *Simple Flake* | 11 | 18 | 29 |
| *Discoid Flake* | | 2 | 2 |
| *Levallois Recurrent Flake* | 18 | 19 | 37 |
| **Total Flakes** | **42** | **63** | **105** |
| **Total Chunks** | **7** | **3** | **10** |
| **Total Fragments** | **12** | **10** | **22** |
| **Total** | **73** | **84** | **157** |

The lithic artefacts are presented in two columns dividing the artefacts from the stratigraphically reliable context from those from the stratigraphically unreliable one. Rows are reporting technological categories (Blades, Bladelets, Flakes, Cores, Chunks, Fragments, in bold) and subcategories (in italics). Plain text below certain subcategories (Core-Edge, Pseudo-Levallois points, Overshot flakes) report the attribution to a Levallois, Discoid or Not assigned method.

followed by calcarenites and fewer examples of radiolarites and quartzite. Northwestern Tuscany is mostly characterised by four palaeogeographic domains: The Ligurian domain, the Subligurian domain, the Tuscan Nappe, and the Tuscan Metamorphic units [57]. In particular, the geological background of Buca della Iena and Grotta del Capriolo consists of the Tuscan Nappe, Subligurian domains and Ligurian domains. The Tuscan Nappe is characterised by marine shelf carbonate sedimentation. In particular, in the area near the sites, the Calcare Cavernoso formation (CCA) is the most common for

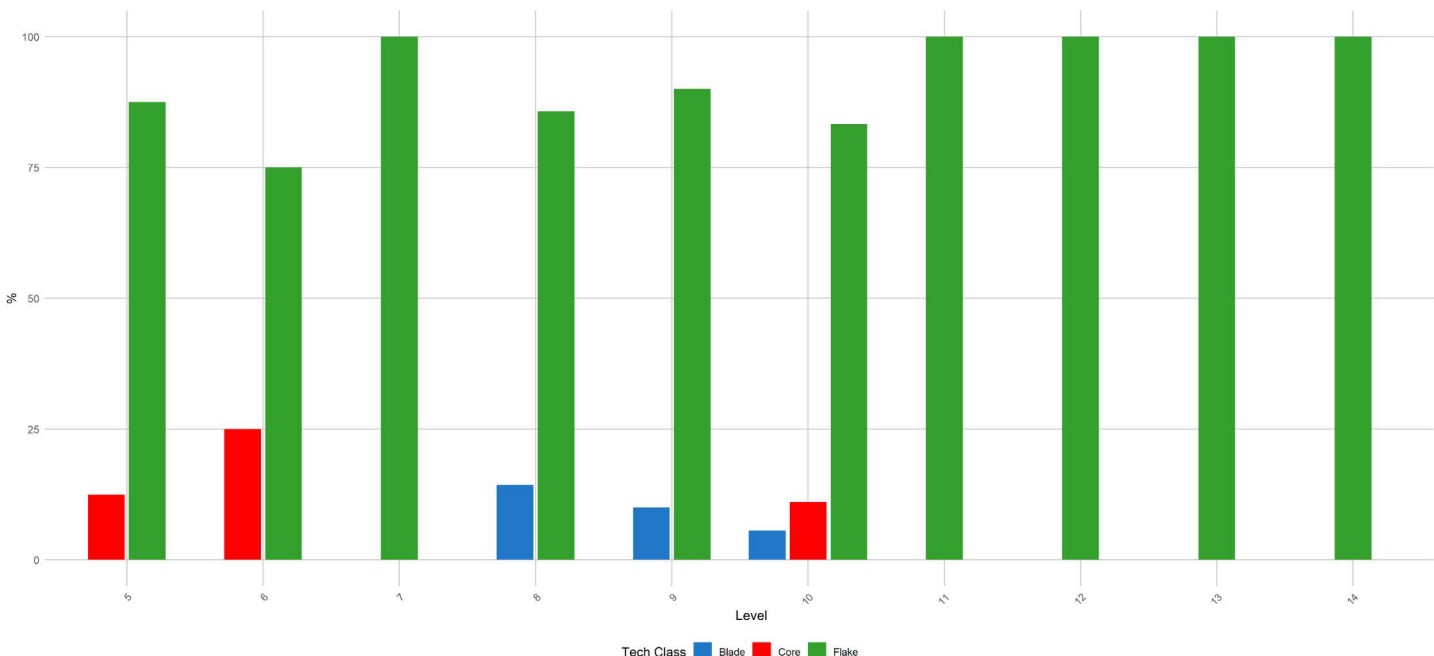

**Fig 9. Buca della lena frequency of blanks throughout the sequence.** The frequency is reported in percentages and pertains only to the stratigraphically reliable assemblage.

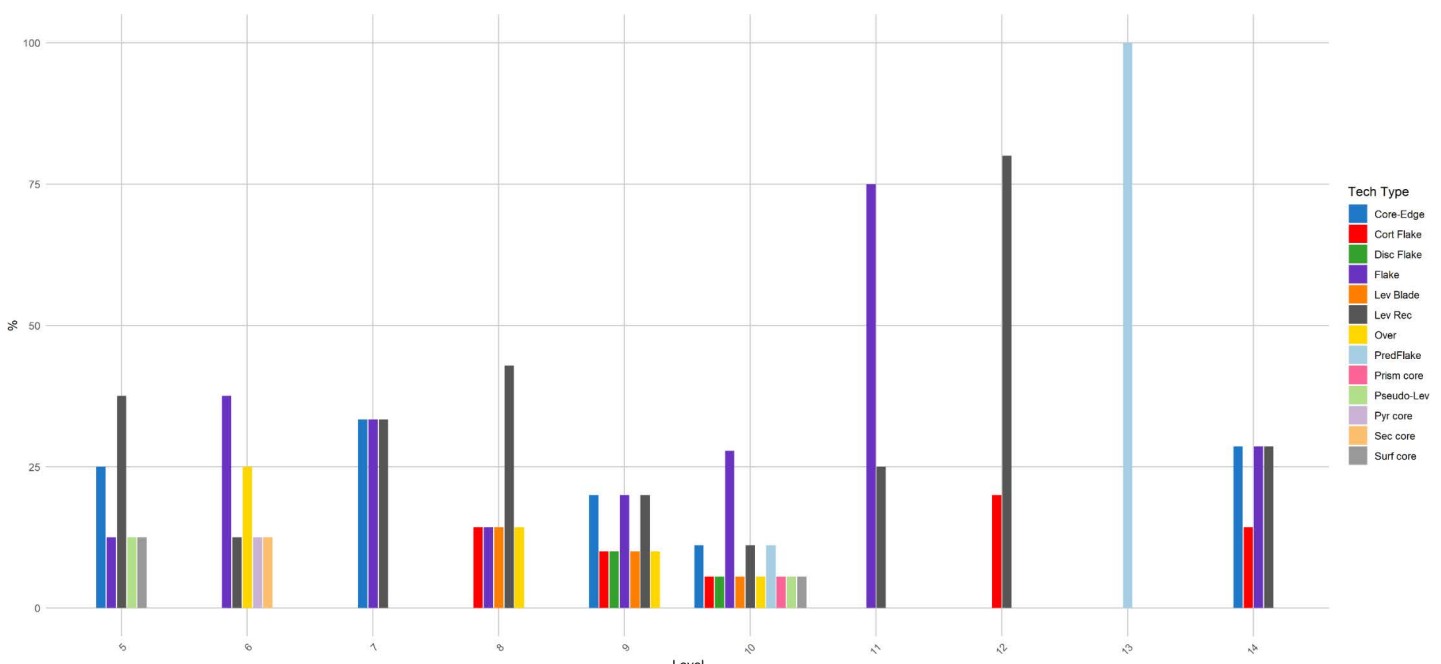

**Fig 10. Buca della lena frequency of technological type throughout the sequence.** The frequency is reported in percentages and pertains only to the stratigraphically reliable assemblage.

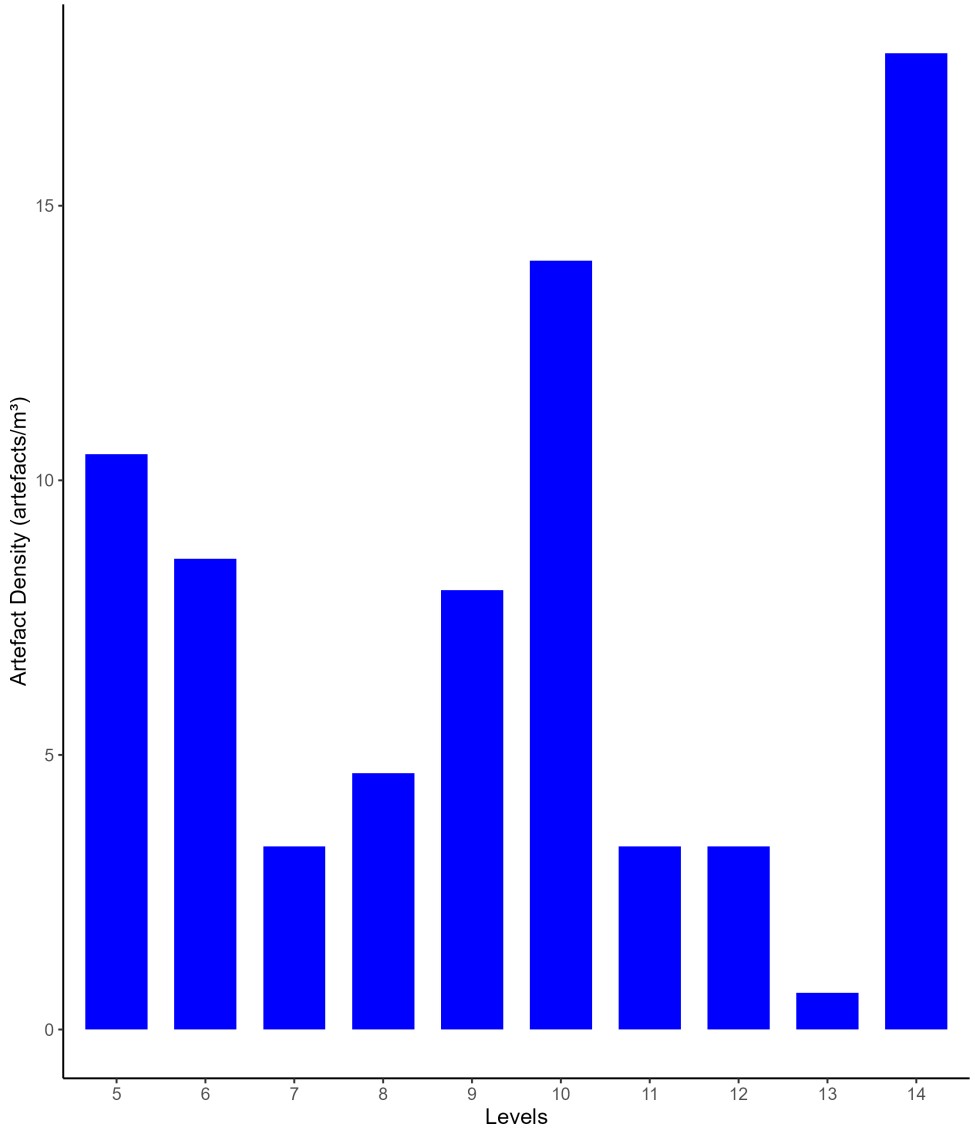

**Fig 11. Buca della lena artefacts densities (number of artefacts/single level volume) throughout the sequence.** It assesses densities only in the stratigraphically reliable sectors.

the formation of karst phenomena, then it is followed by the Calcare Massiccio formation (MAS), and various carbonate formations containing siliceous shales, radiolarites, calcilutites, and calcarenites: Rosso Ammonitico formation (RSA), Calcare selcifero della Val di Lima formation (SVL), Diaspri formation (DSD), Maiolica formation (MAI), Scaglia Toscana formation (STO). It is topped by the Macigno sandstone formation (MAC) [57,58]. The Ligurian domain in the area is characterised by Helminthoid Flysch (OMT - Ottone formation) Campanian-Maastrichtian in age (83.5–65.5 Ma), which is a thick succession (up to 120 m thick) of calcareous turbidites with calcarenite base, marly limestones, thin siliciclastic turbidites and hemipelagic limestones [57,58] The Subligurian domain in the area is characterised the Canetolo Shales and Limestones formation (ACC - up to 200 m thick), which is middle Eocene in age (48–38 Ma), and is characterised by shales, siltstones and limestones. In particular, two facies the ACCb and the ACCa are recognised near the sites,

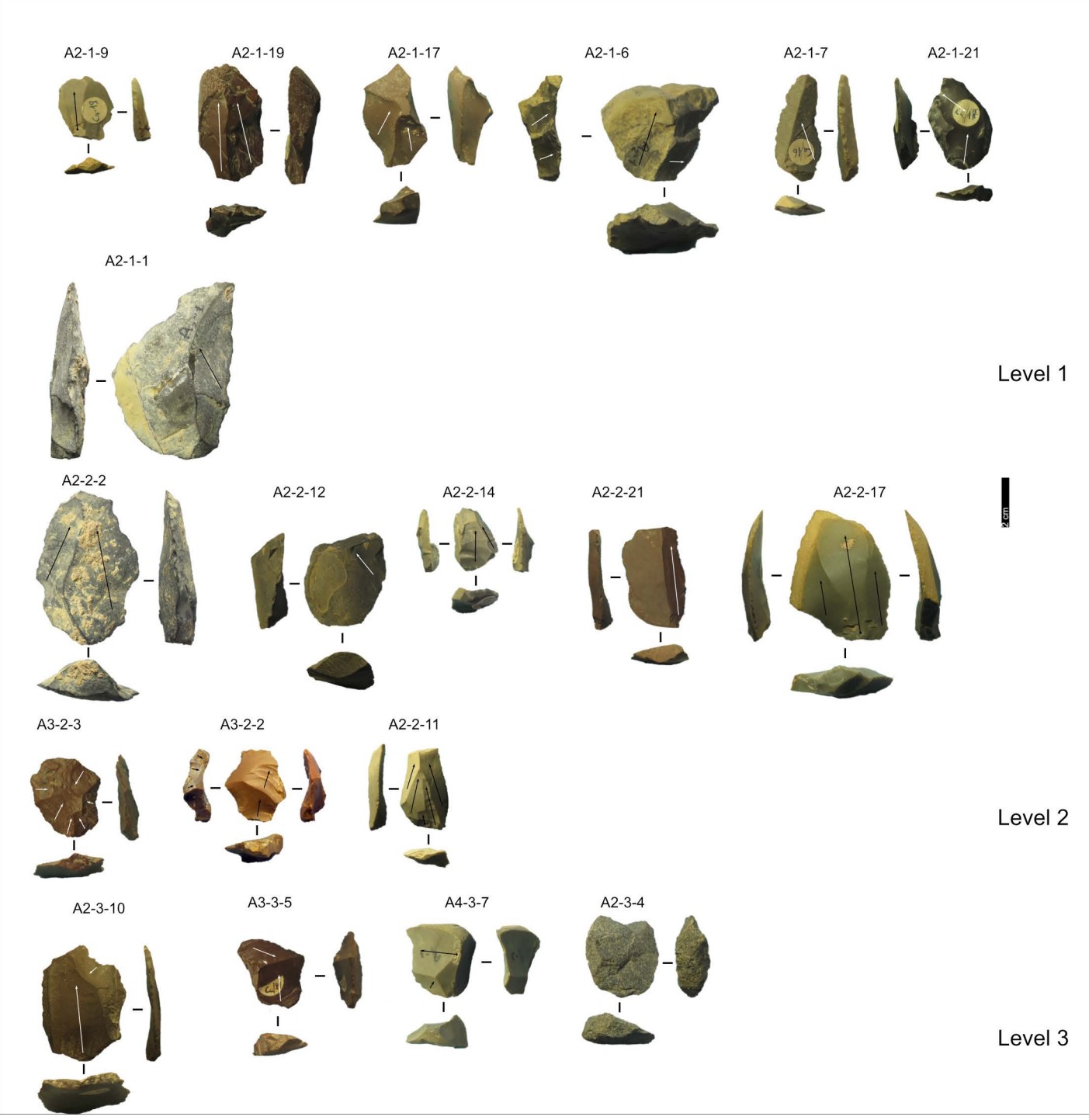

**Fig 12. Grotta del Capriolo lithic artefacts from levels 1 to 3.** IDs above each artefact correspond to IDs reported in S2 and S3 Files. Core-Edge Flake: A2-1-17 (Discoid), A2-1-21, A2-1-7, A2-2-14, A2-2-2, A2-3-4 (Discoid), A3-2-2 (Levallois); Cortical Flake: A2-1-1, A2-2-12, A2-2-21; Levallois point: A2-2-11 Levallois Preferential Flake: A3-2-3; Levallois Recurrent Flake: A2-1-9, A2-2-17, A2-3-10; Overshot Flake: A2-1-19 (Levallois), A2-1-6, A4-3-7 (Discoid); Pseudo-Levallois point: A3-3-5 (Discoid). Pictures Jacopo Gennai.

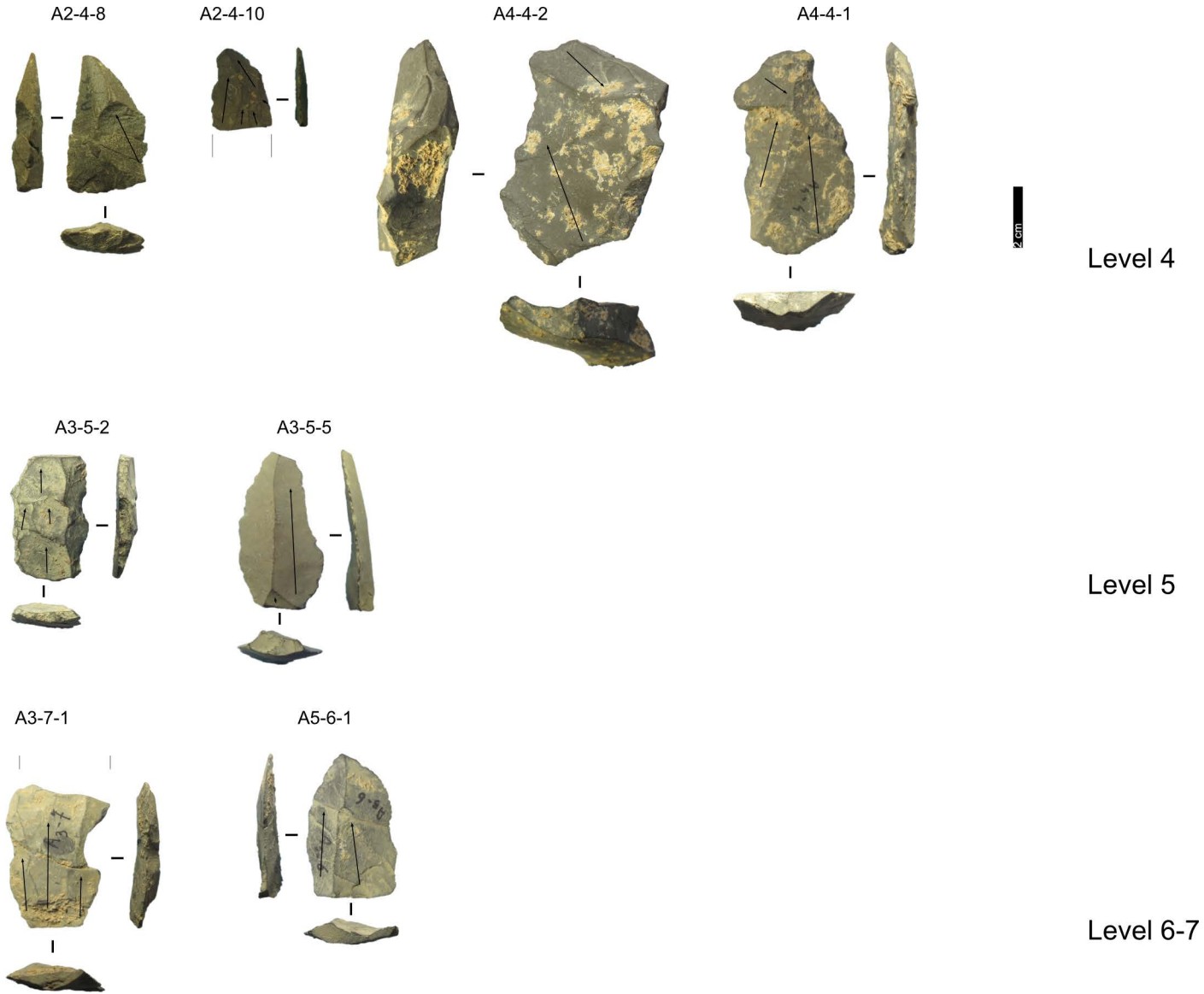

A2-4-8  A2-4-10  A4-4-2  A4-4-1

Level 4

A3-5-2  A3-5-5

Level 5

A3-7-1  A5-6-1

Level 6-7

**Fig 13. Grotta del Capriolo lithic artefacts from levels 4 to 7.** IDs above each artefact correspond to IDs reported in S2 and S3 Files. Core-Edge Flake: A4-4-2; Cortical Flake: A4-4-1; Levallois Blade: A2-4-10, A3-5-2, A3-5-5; Levallois point: A5-6-1; Levallois Recurrent Flake: A2-4-8, A3-7-1. Pictures Jacopo Gennai.

they show dark grey shales, brown-greyish siltstones, calcilutites and grey calcarenites [58]. Surveys have confirmed the presence of calcilutites and calcarenites in the vicinity of the two sites (<5 km), while radiolarite outcrops still need to be found. Observation at the stereomicroscope does not show major signs of rolling of cortical surfaces, hence it is likely that all the raw materials were procured in primary or sub-primary contexts.

## Fauna

**Buca della Iena.** The zooarchaeological analysis at Buca della Iena involved examining 3710 remains of middle—and large-sized mammals recovered from all the levels of sectors C, D, and F.

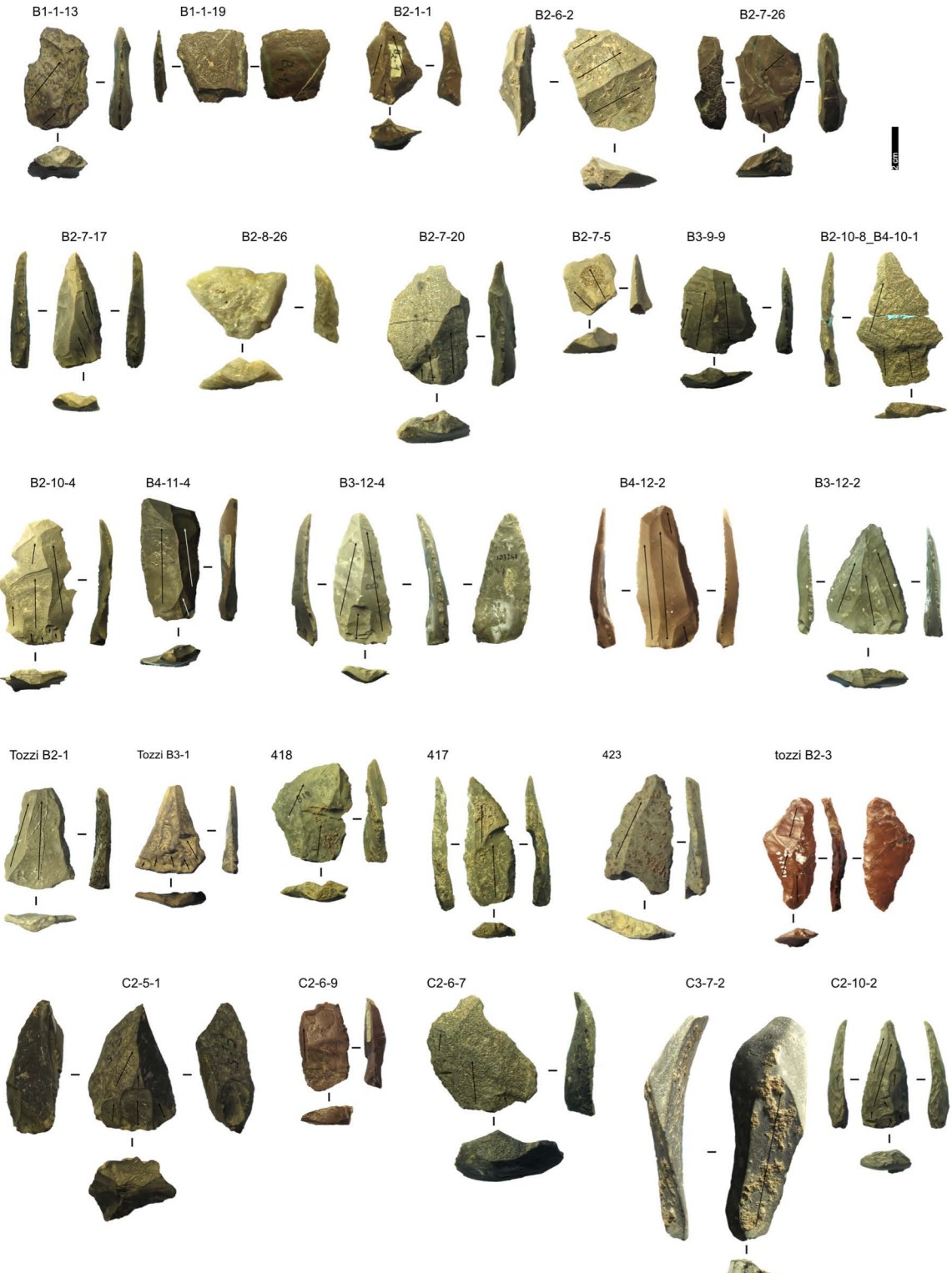

**Fig 14. Grotta del Capriolo lithic artefacts from stratigraphically unreliable context.** IDs above each artefact correspond to IDs reported in S2 and S3 Files. Volumetric Blade: Tozzi B2-3, C3-7-2; Core-Edge Flake: B2-7-26 (Levallois), C2-6-9; Cortical Flake: B1-1-19, B2-7-20, B2-7-5; Discoid Flake: B2-1-1, B2-8-26: Levallois Blade: 417, 423, B2-7-17, B3-12-4, B4-11-4, B4-12-2, C2-10-2, Tozzi B3-1; Levallois point: B3-12-2, Tozzi B2-1; Levallois Recurrent Flake: B1-1-13, B2-10-4＋B2-10-8, B3-9-9; Predetermining Flake: 418; Pseudo-Levallois point: B2-6-2 (Levallois). Pictures Jacopo Gennai.

**Table 3. Grotta del Capriolo lithic artefacts categories and counts.**

| Grotta del Capriolo | Stratigraphically unreliable | Stratigraphically reliable | Total |
|---|---|---|---|
| **Blade** | | | |
| *Volumetric Blade* | 9 | 5 | 14 |
| *Core-Edge* | 18 | 3 | 21 |
| Discoid concept | 1 | | 1 |
| Levallois Concept | 4 | 1 | 5 |
| Not assigned | 13 | 2 | 15 |
| *Cortical* | 5 | | 5 |
| *Laminar Asymmetrical* | 3 | | 3 |
| *Levallois Blade* | 22 | 7 | 29 |
| *Levallois Point* | 1 | 1 | 2 |
| *Levallois Recurrent Elongated Flake* | 2 | | 2 |
| *Overshot* | 8 | | 8 |
| Levallois Concept | 3 | | 3 |
| Not assigned | 5 | | 5 |
| **Total Blades** | **68** | **16** | **84** |
| **Core** | | | |
| *Centripetal* | 1 | | 1 |
| *Secant unifacial core* | 1 | | 1 |
| *Non organised globular* | 1 | 1 | 2 |
| *Pyramidal* | 1 | | 1 |
| *Secant* | 1 | | 1 |
| *Surface* | 1 | | 1 |
| **Total Cores** | **6** | **1** | **7** |
| **Flake** | | | |
| *Core-Edge Flake* | 51 | 13 | 64 |
| Discoid concept | 14 | 5 | 19 |
| Levallois Concept | 9 | 1 | 10 |
| Not assigned | 28 | 7 | 35 |
| *Cortical Flake* | 45 | 10 | 55 |
| *Crest* | | 1 | 1 |
| *Discoid Flake* | 26 | 6 | 32 |
| *Flake* | 78 | 21 | 99 |
| *Kombewa flake* | 1 | | 1 |
| *Levallois Point* | 7 | | 7 |
| *Levallois Preferential Flake* | 3 | 1 | 4 |
| *Levallois Recurrent Flake* | 62 | 12 | 74 |
| *Overshot Flake* | 25 | 2 | 27 |
| Discoid concept | 4 | | 4 |
| Levallois Concept | 7 | 1 | 8 |
| Not assigned | 14 | 1 | 15 |
| *Predetermining Flake* | 11 | 3 | 14 |
| Levallois Concept | 2 | | 2 |
| Not assigned | 9 | 3 | 12 |
| *Pseudo-Levallois point* | 27 | 4 | 31 |
| Not assigned | 10 | | 10 |
| Discoid concept | 10 | 2 | 12 |

*(Continued)*

## PLOS One

**Table 3.** (Continued)

| Grotta del Capriolo | Stratigraphically unreliable | Stratigraphically reliable | Total |
|---|---|---|---|
| Levallois Concept | 7 | 2 | 9 |
| **Total Flakes** | **336** | **73** | **409** |
| **Total Fragments** | **30** | **9** | **39** |
| **Total** | **440** | **99** | **539** |

The lithic artefacts are presented in two columns dividing the artefacts from the stratigraphically reliable context from those from the stratigraphically unreliable one. Rows are reporting technological categories (Blades, Bladelets, Flakes, Cores, Chunks, Fragments, in bold) and subcategories (in italics). Plain text below certain subcategories (Core-Edge, Pseudo-Levallois points, Overshot flakes) report the attribution to a Levallois, Discoid or Not assigned method.

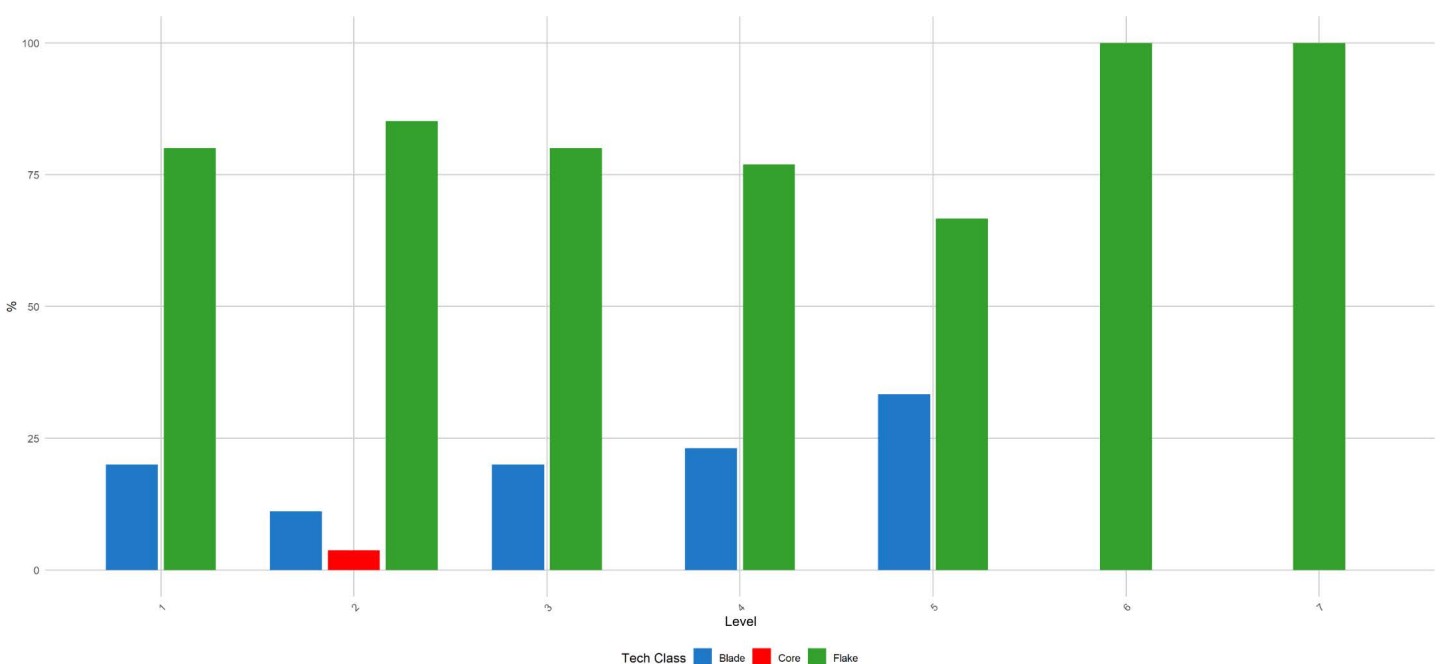

**Fig 15. Grotta del Capriolo frequency of blanks throughout the sequence.** The frequency is reported in percentages and pertains only to the stratigraphically reliable assemblage.

**Taxonomy.** Approximately 19% of the faunal remains (N = 683) are identifiable and they reveal the presence of at least 18 distinct *taxa*. These include *Erinaceus europaeus*, *Lepus* sp., *Marmota marmota, Canis lupus*, *Ursus spelaeus*, *Ursus arctos*, *Meles meles, Mustela* sp., *Panthera pardus*, *Crocuta spelaea*, *Palaeoloxodon antiquus*, *Rhinocerotidae*, *Equus ferus*, *Sus scrofa*, *Capreolus capreolus*, *Megaloceros giganteus*, *Cervus elaphus*, and *Bos primigenius* (Table 4).

However, 78% of the specimens (N = 2852) do not show taxonomically determinable characteristics. Additionally, vertebral fragments (N = 27) and rib fragments (N = 96), which could not be taxonomically attributed (3% overall), were sorted by size into three categories: small, medium, and large (Table 4).

*U. spelaeus* is the most identified species across all stratigraphic levels at Buca della Iena, with the exception of Level 15, where *C. spelaea* predominates. The latter species is present in every spit, along with the main ungulate species. The badger (*M. meles*) is primarily found in the upper levels, while it is nearly absent in the lower ones (levels 13 and 14). Remains of wolf (*C. lupus*), mustelid and leopard (*P. pardus*) are recorded in some levels but are relatively rare. Regarding

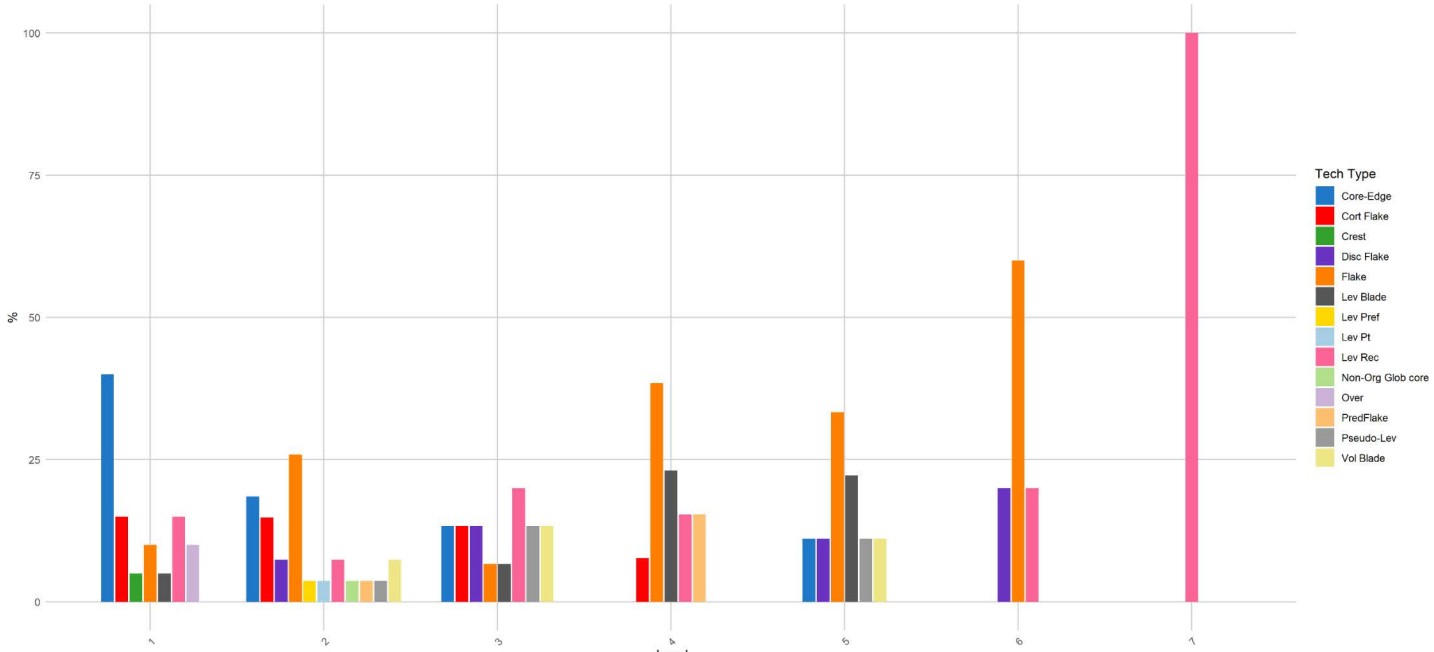

**Fig 16. Grotta del Capriolo frequency of technological type throughout the sequence.** The frequency is reported in percentages and pertains only to the stratigraphically reliable assemblage.

the frequency distribution of ungulate species, red deer (*C. elaphus*) and horse (*E. ferus*) consistently co-dominate across all levels, with the aurochs (*B. primigenius*) being the next most common species in the majority of Levels. Notably, horse remains are particularly abundant in Level 15, where a substantial number of teeth have been recovered. In contrast, lagomorphs and rodents are scarce throughout the sequence (Table 4).

According to the MNI, the most common ungulates are *E. ferus* and *C. elaphus*, with a predominance of adult individuals in each stratigraphical level, followed by very young individuals. In contrast, senile and young individuals are represented in much smaller numbers (Table 5). The same trend is observed for *U. spelaeus* and *C. spelaea*, the most common carnivores. In Level 15, the number of very young and young individuals of the latter increases, leading to the interpretation of this deposit as a proper hyena den. For the other species, a prevalence of adults is observed (Table 5).

**Taphonomy.** Taphonomic modifications on the surfaces of bones are recorded in various degrees across the sequence. In general, lower levels record a higher frequency of specimens with modification, reaching the maximum in level 15 (74% - Fig 18 and S4 File).

In each level, more than 66% of the observed traces can be attributed to cave hyena activity on the carcasses (see Figs 19 and 20, and S5 File). This evidence includes mainly fragmented diaphyses with cracks characteristic of carnivore activity and puncture marks, morphometrically attributed to cave hyena's bites, widespread scoring, underrepresented epiphyses with furrows, digested fragments and nibbling sticks made by hyena cubs. These modifications suggest that cave hyenas were the primary agents of bone damage, especially in level 15.

Human activity in the cave appears to have been sporadic, with a low percentage of modifications, ranging from 0% to 3.2% (see Fig 19). Cut marks and butchery marks were identified on a hyena radius from level 5, two red deer metatarsals from level 15, and several indeterminate bone fragments from levels 7, 9, 10, 14, and 15 (Fig 20).

Additional modifications include rodent gnawing marks (0–3.6%), small carnivore gnawing (0–1.3%), ichnotraces (0–0.2%), root etching (0–1.8%), and mineral manganese patinas covering between 8–33.3% of the bones (Fig 20 and S5 File).

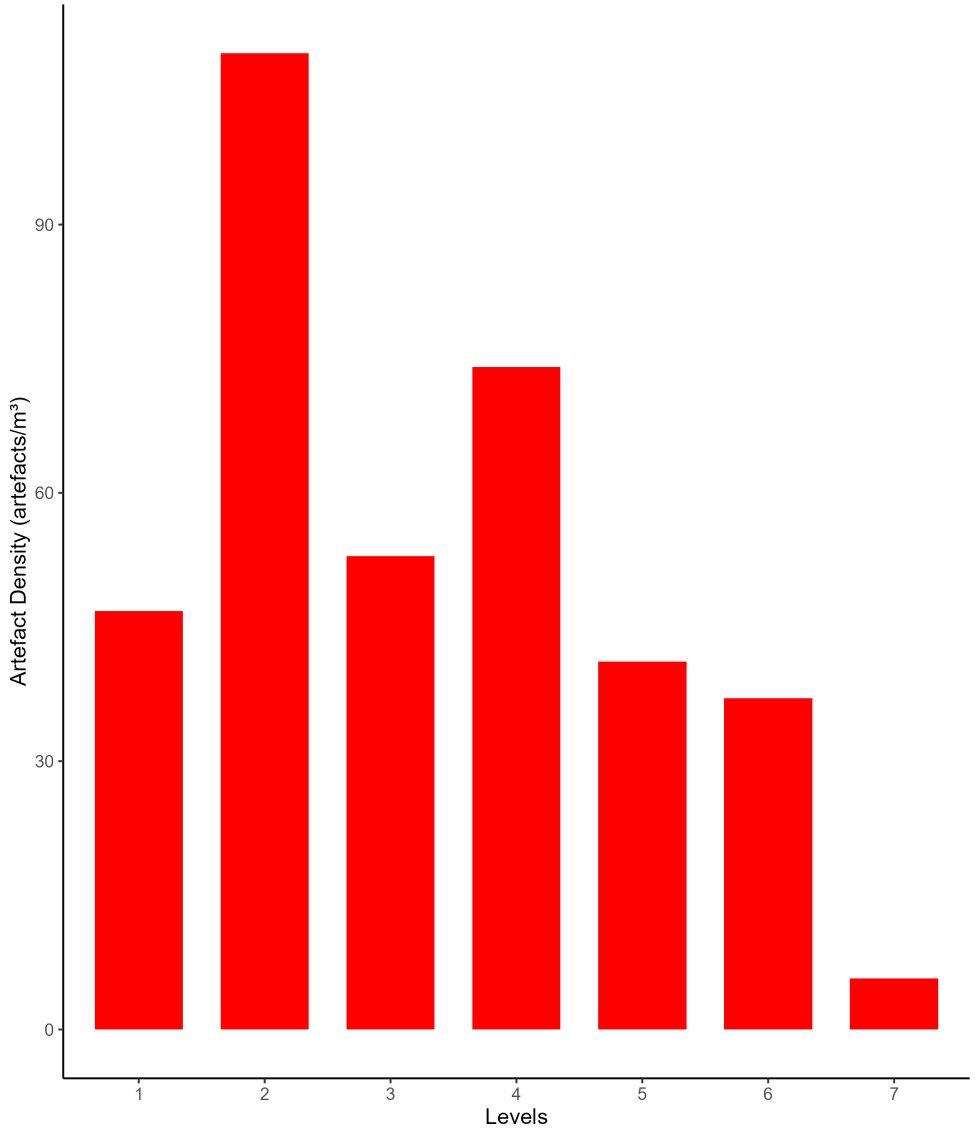

**Fig 17. Grotta del Capriolo artefacts densities (number of artefacts/volume of levels) throughout the sequence.** It assesses only stratigraphically reliable sectors.

## Grotta del Capriolo

The zooarchaeological study focused on 273 selected remains of middle- and large-sized mammals from areas A2 and A3 of Grotta del Capriolo.

**Taxonomy.** Approximately 4% of the total finds (N = 11) are taxonomically identifiable at the Grotta del Capriolo, representing at least six distinct *taxa*. These include *Felis silvestris*, *Ursus* sp., *Rhinocerotidae*, *Equus ferus*, *Sus scrofa*, and *Capreolus capreolus* (Table 6). The majority of the material (262 bones, which is about 96% of the total) is too fragmented or eroded to allow for taxonomic identification and is therefore classified as indeterminate. The limited number of fragments that can be identified both taxonomically and anatomically does not allow for further insights into the species that were present.

**Table 4. NISP from each level of Buca della Iena.**

| | Level 5 | | Level 6 | | Level 7 | | Level 8 | | Level 9 | | Level 10 | | Level 11 | | Level 12 | | Level 13 | | Level 14 | | Level 15 | | Total remains | |
|---|---|---|---|---|---|---|---|---|---|---|---|---|---|---|---|---|---|---|---|---|---|---|---|---|
| | NISP | % | NISP | % | NISP | % | NISP | % | NISP | % | NISP | % | NISP | % | NISP | % | NISP | % | NISP | % | NISP | % | NISP | % |
| Erinaceus europaeus | 1 | 2,2% | 0 | 0,0% | 0 | 0,0% | 0 | 0,0% | 0 | 0,0% | 0 | 0,0% | 0 | 0,0% | 0 | 0,0% | 0 | 0,0% | 0 | 0,0% | 0 | 0,0% | 1 | 0,1% |
| Lepus sp. | 0 | 0,0% | 0 | 0,0% | 0 | 0,0% | 1 | 2,9% | 0 | 0,0% | 0 | 0,0% | 0 | 0,0% | 0 | 0,0% | 0 | 0,0% | 0 | 0,0% | 1 | 0,2% | 2 | 0,3% |
| Marmota marmota | 1 | 2,2% | 0 | 0,0% | 0 | 0,0% | 0 | 0,0% | 0 | 0,0% | 0 | 0,0% | 0 | 0,0% | 0 | 0,0% | 0 | 0,0% | 0 | 0,0% | 0 | 0,0% | 1 | 0,1% |
| Canis lupus | 1 | 2,2% | 1 | 5,6% | 0 | 0,0% | 0 | 0,0% | 1 | 2,6% | 0 | 0,0% | 1 | 4,2% | 2 | 15,4% | 0 | 0,0% | 5 | 15,6% | 1 | 0,2% | 12 | 1,7% |
| Ursus spelaeus | 17 | 37,0% | 6 | 33,3% | 17 | 63,0% | 21 | 61,8% | 18 | 47,4% | 20 | 50,0% | 13 | 54,2% | 6 | 46,2% | 6 | 75,0% | 20 | 62,5% | 27 | 6,5% | 171 | 24,5% |
| Ursus arctos | 1 | 2,2% | 0 | 0,0% | 0 | 0,0% | 0 | 0,0% | 0 | 0,0% | 0 | 0,0% | 0 | 0,0% | 0 | 0,0% | 0 | 0,0% | 0 | 0,0% | 0 | 0,0% | 1 | 0,1% |
| Meles meles | 5 | 10,9% | 5 | 27,8% | 2 | 7,4% | 4 | 11,8% | 4 | 10,5% | 6 | 15,0% | 2 | 8,3% | 1 | 7,7% | 0 | 0,0% | 0 | 0,0% | 1 | 0,2% | 30 | 4,3% |
| Mustela sp. | 0 | 0,0% | 0 | 0,0% | 1 | 3,7% | 0 | 0,0% | 0 | 0,0% | 0 | 0,0% | 0 | 0,0% | 0 | 0,0% | 0 | 0,0% | 0 | 0,0% | 0 | 0,0% | 1 | 0,1% |
| Panthera pardus | 0 | 0,0% | 0 | 0,0% | 0 | 0,0% | 0 | 0,0% | 2 | 5,3% | 0 | 0,0% | 0 | 0,0% | 0 | 0,0% | 0 | 0,0% | 0 | 0,0% | 0 | 0,0% | 2 | 0,3% |
| Crocuta spelaea | 11 | 23,9% | 6 | 33,3% | 4 | 14,8% | 4 | 11,8% | 4 | 10,5% | 4 | 10,0% | 3 | 12,5% | 1 | 7,7% | 1 | 12,5% | 4 | 12,5% | 115 | 27,5% | 157 | 22,5% |
| Palaeoloxodon antiquus | 0 | 0,0% | 0 | 0,0% | 0 | 0,0% | 0 | 0,0% | 0 | 0,0% | 0 | 0,0% | 0 | 0,0% | 0 | 0,0% | 0 | 0,0% | 0 | 0,0% | 1 | 0,2% | 1 | 0,1% |
| Rhinocerotidae | 1 | 2,2% | 0 | 0,0% | 0 | 0,0% | 0 | 0,0% | 0 | 0,0% | 1 | 2,5% | 0 | 0,0% | 0 | 0,0% | 0 | 0,0% | 0 | 0,0% | 4 | 1,0% | 6 | 0,9% |
| Equus ferus | 1 | 2,2% | 0 | 0,0% | 2 | 7,4% | 0 | 0,0% | 2 | 5,3% | 1 | 2,5% | 0 | 0,0% | 1 | 7,7% | 1 | 12,5% | 1 | 3,1% | 119 | 28,5% | 128 | 18,3% |
| Sus scrofa | 2 | 4,3% | 0 | 0,0% | 0 | 0,0% | 1 | 2,9% | 2 | 5,3% | 1 | 2,5% | 0 | 0,0% | 0 | 0,0% | 0 | 0,0% | 0 | 0,0% | 2 | 0,5% | 8 | 1,1% |
| Capreolus capreolus | 2 | 4,3% | 0 | 0,0% | 1 | 3,7% | 2 | 5,9% | 0 | 0,0% | 3 | 7,5% | 1 | 4,2% | 2 | 15,4% | 0 | 0,0% | 0 | 0,0% | 3 | 0,7% | 14 | 2,0% |
| Megaloceros giganteus | 1 | 2,2% | 0 | 0,0% | 0 | 0,0% | 0 | 0,0% | 0 | 0,0% | 0 | 0,0% | 1 | 4,2% | 0 | 0,0% | 0 | 0,0% | 0 | 0,0% | 4 | 1,0% | 6 | 0,9% |
| Cervus elaphus | 0 | 0,0% | 0 | 0,0% | 0 | 0,0% | 1 | 2,9% | 5 | 13,2% | 4 | 10,0% | 1 | 4,2% | 0 | 0,0% | 0 | 0,0% | 1 | 3,1% | 102 | 24,4% | 114 | 16,3% |
| Bos primigenius | 2 | 4,3% | 0 | 0,0% | 0 | 0,0% | 0 | 0,0% | 0 | 0,0% | 0 | 0,0% | 2 | 8,3% | 0 | 0,0% | 0 | 0,0% | 1 | 3,1% | 38 | 9,1% | 43 | 6,2% |

*(Continued)*

**Table 4.** (Continued)

| | Level 5 NISP | % | Level 6 NISP | % | Level 7 NISP | % | Level 8 NISP | % | Level 9 NISP | % | Level 10 NISP | % | Level 11 NISP | % | Level 12 NISP | % | Level 13 NISP | % | Level 14 NISP | % | Level 15 NISP | % | Total remains NISP | % |
|---|---|---|---|---|---|---|---|---|---|---|---|---|---|---|---|---|---|---|---|---|---|---|---|---|
| **Total identified specimens** | 46 | 100% | 18 | 100% | 27 | 100% | 34 | 100% | 38 | 100% | 40 | 100% | 24 | 100% | 13 | 100% | 8 | 100% | 32 | 100% | 418 | 100% | 698 | 100% |
| Vertebrae S | 1 | 0,4% | 0 | 0,0% | 0 | 0,0% | 0 | 0,0% | 1 | 0,3% | 0 | 0,0% | 0 | 0,0% | 0 | 0,0% | 0 | 0,0% | 0 | 0,0% | 1 | 0,1% | 3 | 0,1% |
| Vertebrae M | 1 | 0,4% | 1 | 0,8% | 2 | 0,8% | 0 | 0,0% | 0 | 0,0% | 1 | 0,4% | 2 | 1,1% | 0 | 0,0% | 0 | 0,0% | 0 | 0,0% | 9 | 0,5% | 16 | 0,4% |
| Vertebrae L | 0 | 0,0% | 0 | 0,0% | 2 | 0,8% | 0 | 0,0% | 1 | 0,3% | 1 | 0,4% | 0 | 0,0% | 1 | 0,9% | 0 | 0,0% | 0 | 0,0% | 3 | 0,2% | 8 | 0,2% |
| Ribs S | 2 | 0,7% | 3 | 2,3% | 0 | 0,0% | 4 | 1,9% | 3 | 0,9% | 0 | 0,0% | 0 | 0,0% | 0 | 0,0% | 0 | 0,0% | 2 | 1,4% | 0 | 0,0% | 14 | 0,4% |
| Ribs M | 4 | 1,4% | 2 | 1,6% | 3 | 1,2% | 2 | 1,0% | 1 | 0,3% | 1 | 0,4% | 2 | 1,1% | 2 | 1,9% | 0 | 0,0% | 6 | 4,1% | 19 | 1,1% | 42 | 1,1% |
| Ribs L | 2 | 0,7% | 0 | 0,0% | 4 | 1,6% | 1 | 0,5% | 2 | 0,6% | 0 | 0,0% | 3 | 1,6% | 2 | 1,9% | 1 | 1,8% | 1 | 0,7% | 24 | 1,4% | 40 | 1,1% |
| Indeterminate bones | 224 | 80,0% | 105 | 81,4% | 208 | 84,6% | 166 | 80,2% | 278 | 85,8% | 208 | 82,9% | 156 | 83,4% | 90 | 83,3% | 46 | 83,6% | 107 | 72,3% | 1301 | 73,3% | 2889 | 77,9% |
| **Total remains** | 280 | 100% | 129 | 100% | 246 | 100% | 207 | 100% | 324 | 100% | 251 | 100% | 187 | 100% | 108 | 100% | 55 | 100% | 148 | 100% | 1775 | 100% | 3710 | 100% |

**Table 5. MNI from each level of Buca della Iena.**

| Species | L5 YY | L5 Y | L5 A | L5 S | L6 YY | L6 Y | L6 A | L6 S | L7 YY | L7 Y | L7 A | L7 S | L8 YY | L8 Y | L8 A | L8 S | L9 YY | L9 Y | L9 A | L9 S | L10 YY | L10 Y | L10 A | L10 S | L11 YY | L11 Y | L11 A | L11 S | L12 YY | L12 Y | L12 A | L12 S | L13 YY | L13 Y | L13 A | L13 S | L14 YY | L14 Y | L14 A | L14 S | L15 YY | L15 Y | L15 A | L15 S |
|---|---|---|---|---|---|---|---|---|---|---|---|---|---|---|---|---|---|---|---|---|---|---|---|---|---|---|---|---|---|---|---|---|---|---|---|---|---|---|---|---|---|---|---|---|
| *Erinaceus europaeus* | | | 1 | | | | | | | | | | | | | | | | | | | | | | | | | | | | | | | | | | | | | | | | | |
| *Lepus* sp. | | | | | | | | | | | | | | | 1 | | | | | | | | | | | | | | | | | | | | | | | | | | | 1 | | |
| *Marmota marmota* | | | 1 | | | | | | | | | | | | | | | | | | | | | | | | | | | | | | | | | | | | | | | | | |
| *Canis lupus* | | | 1 | | | | 1 | | | | | | | | | | | | 1 | | | | | | | | 1 | | | | 1 | | | | | | | | 1 | | | | 1 | |
| *Ursus spelaeus* | | 1 | 1 | 1 | | 1 | 1 | 1 | | | 2 | | 1 | 1 | 2 | 1 | 1 | | 2 | | | 1 | 3 | 1 | | | 2 | 2 | | | 1 | | | | 2 | 1 | | 1 | 2 | 1 | | 3 | 3 | 2 |
| *Ursus arctos* | | | 1 | | | | | | | | | | | | | | | | | | | | | | | | | | | | | | | | | | | | | | | | | |
| *Meles meles* | | | 1 | | | | 1 | | | | 1 | | | | 1 | | | | 2 | | | 1 | 1 | | | | 1 | | 1 | | | | | | | | | | | | | | 1 | |
| *Mustela* sp. | | | | | | | | | | | 1 | | | | | | | | | | | | | | | | | | | | | | | | | | | | | | | | | |
| *Panthera pardus* | | | | | | | | | | | | | | | | | | | 1 | | | | | | | | | | | | | | | | | | | | | | | | | |
| *Crocuta spelaea* | | | 3 | | | 1 | 2 | | | | 1 | | | 1 | 1 | | | 1 | 1 | | | 1 | 1 | | | | 1 | | | | | | 1 | | 1 | | | | 1 | | 6 | 1 | 7 | 1 |
| *Palaeoloxodon antiquus* | | | | | | | | | | | | | | | | | | | | | | | | | | | | | | | | | | | | | | | | | | | | |
| Rhinocerotidae | | | 1 | | | | | | | | | | | | | | | | | | | | 1 | | | | | | | | | | | | | | | | | | | 1 | | |
| *Equus ferus* | | | 1 | | | | | | | | 1 | | | | | | | | 1 | 1 | 1 | | | | | | | | | | 1 | | | | 1 | | | | 1 | | | 2 | 7 | 2 |
| *Sus scrofa* | | | 1 | | | | | | | | | | | | 1 | | | | 2 | | | | | 1 | | | | | | | | | | | | | | | | | | | 2 | |
| *Capreolus capreolus* | | | 1 | | | | | | | | 1 | | | | 1 | | | | | | | | 1 | | | | 1 | | | | 1 | | | | | | | | | | | | 1 | |
| *Megaloceros giganteus* | | | 1 | | | | | | | | | | | | | | | | | | | | | | | | 1 | | | | 1 | | | | | | | | | | | | 1 | |
| *Cervus elaphus* | | | | | | | | | | | | | | | 1 | | | 1 | 1 | | | | 1 | | | | 1 | | | | | | | | | | | | 1 | | 5 | | 5 | 2 |
| *Bos primigenius* | 1 | | | | | | | | | | | | | | | | | | | | | | | | | | | | | | | | | | | | | | 1 | | 1 | | 5 | |
| **Total** | 0 | 2 | 14 | 1 | 0 | 2 | 5 | 1 | 0 | 0 | 7 | 0 | 2 | 1 | 8 | 1 | 0 | 3 | 11 | 1 | 2 | 2 | 8 | 2 | 1 | 0 | 9 | 2 | 2 | 0 | 4 | 1 | 1 | 0 | 3 | 1 | 0 | 1 | 7 | 0 | 12 | 4 | 36 | 8 |
| | **17** | | | | **8** | | | | **7** | | | | **12** | | | | **15** | | | | **14** | | | | **12** | | | | **7** | | | | **5** | | | | **8** | | | | **60** | | | |

YY = Very Young
Y = Young
A = Adults
S = senile

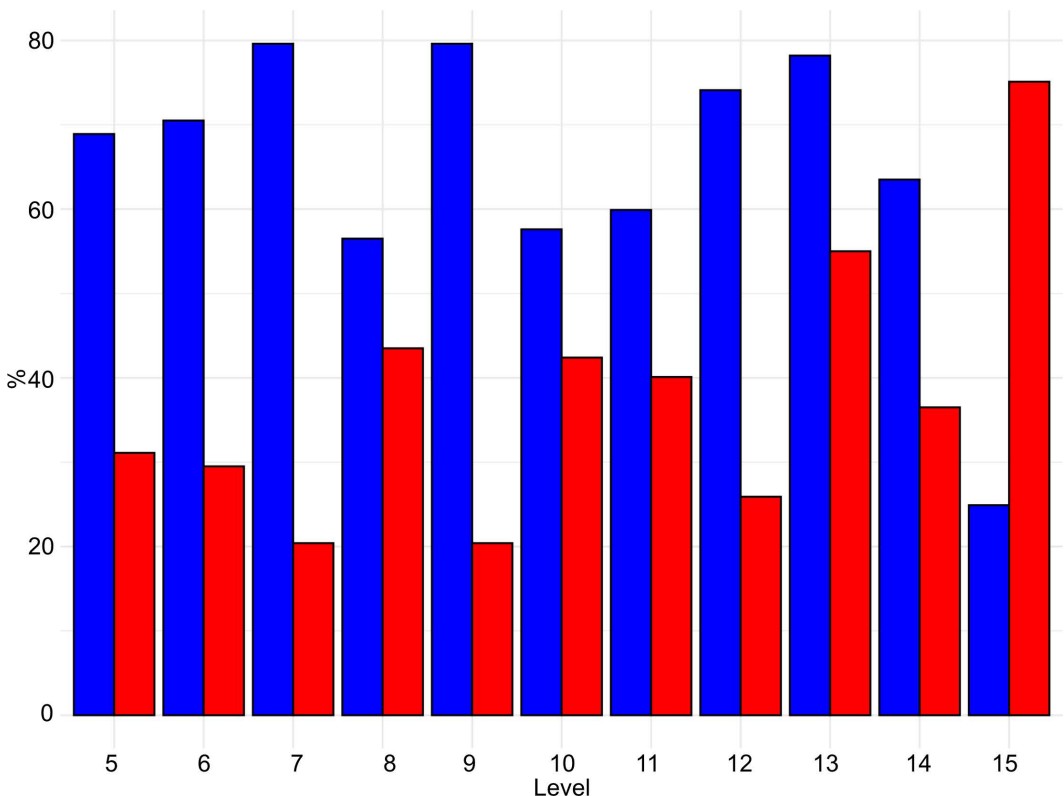

**Fig 18. Buca della lena frequency of bones with taphonomical traces (red) and without taphonomical traces (blue).** Data is expressed in percentage and organised according to stratigraphical levels.

**Taphonomy.** The taphonomical analysis of the faunal assemblage from Grotta del Capriolo reveals that about 45% of the bones has surface modifications (Figs 21–23 and S6 File). In area A2, 52 bones (representing 43% of the total) show traces attributed to various agents (S7 File): human activity (2 cut marks from level A2-4), small carnivores (3 gnawing marks from level A2-2), and environmental factors, such as the formation of mineral coatings (89 manganese patinas).

Similarly, in area A3, a total of 98 bones (62% of the total) exhibit traces (S7 File). These include evidence of human activity (1 cut mark from A3-4), small carnivore activity (8 gnawing marks), and 89 bones showing manganese patinas.

## Radiocarbon dating

Collagen extraction was performed on 6 samples from Buca della lena and 4 samples from Grotta del Capriolo for radiocarbon dating and stable isotope analysis (Table 7). The expected collagen preservation, based on the nitrogen content (%N), revealed variation in the degree of bone preservation between samples. Several quality criteria have been established to ensure high quality and reliable isotope data. Extracted collagen should have an atomic C:N ratio between 2.9 and 3.6 [104], although in the Higham Laboratory our range is 2.9–3.5 (*c.f.* [105]), while collagen yields ought to be > 1%, or >0.5% when using ultrafiltration approaches. All samples produced excellent C:N ratios, although several samples had collagen yields that were lower than ideal (R00256.1/UCIAMS-286279, R00259.1/UCIAMS-286507, R00262.1/UCIAMS-286508 and R00271.1/UCIAMS-286509). These were radiocarbon dated, but to test whether the results were accurate, we also undertook a second collagen extraction with a larger starting mass of bone in an attempt to increase the collagen yields and assess accuracy. This was done for four samples (marked with subsample numbers of.2). In three cases, this resulted in both increased collagen yields and older $^{14}C$ ages, which were in better agreement with

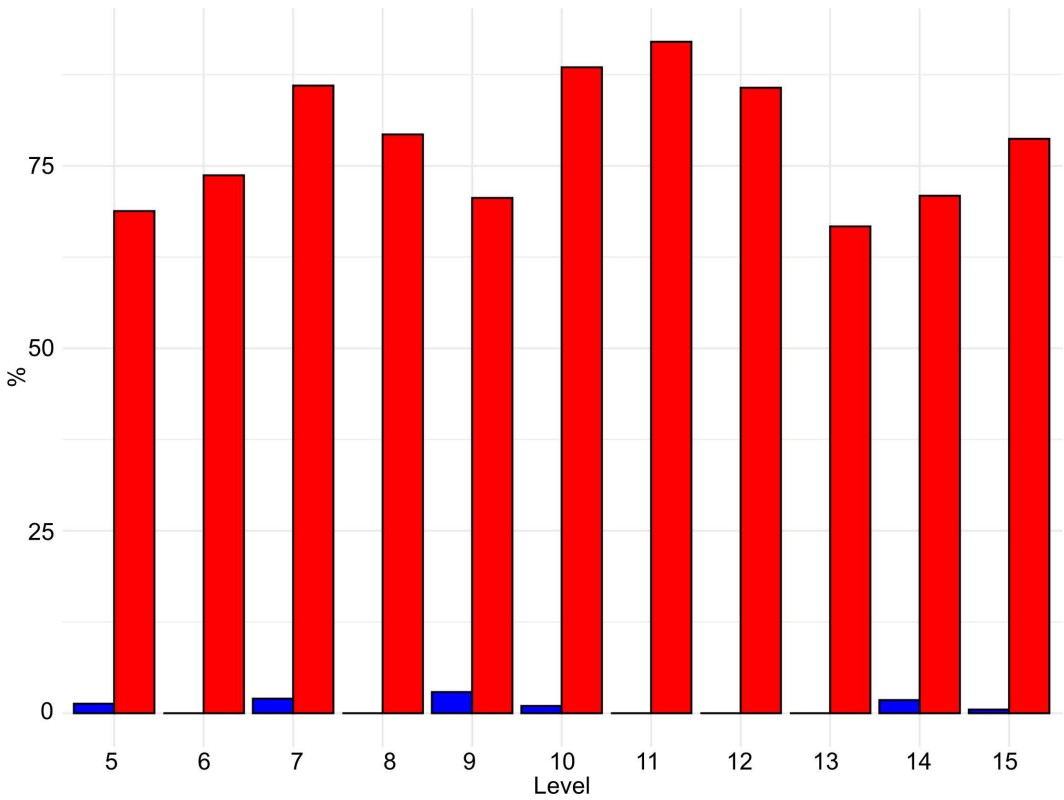

**Fig 19. Buca della lena frequency of human (blue) and hyaena (red) identified alterations on bones.** Data is expressed in percentage and organised according to stratigraphical levels.

the chronological sequence. One sample (R00271.2) produced the same yield as the first extraction, after which it was decided to not continue with radiocarbon dating for this sample. The $^{14}$C age obtained for R00271.1/UCIAMS-286509 should, therefore, be regarded as a minimum age. This comparison shows once again that for poorly preserved material, it is essential to adhere strictly to the quality parameters to obtain reliable radiocarbon results and not underestimates.

We built an age model for the Buca della lena site using OxCal 4.4 [106] and the INTCAL20 dataset [107] (Fig 24). We used a `General` outlier model and constructed the age model using the various Levels and phases used in the excavation. The AMS determinations were included in the model in Fraction Modern (fM) notation to account for asymmetry in the error terms associated with conventional ages (BP) near the limit of the radiocarbon method. We added a previously obtained AMS measurement of >47600 BP (OxA-X-3211-32) from the basal contexts of the site. This result was given an OxA-X- prefix, suggesting it has some potential issues in terms of its reliability, however, it is a minimum age, so informative in this respect. We used a code in OxCal that allows us to include it as a greater than calendar age (the `CQL` Code for the model is given in the Supplementary Information). The model was quick to converge, and convergence values were high. Repeat runs of the model disclosed little variability.

The results of the outlier detection suggest that there are two outliers of significance; R00256.2 (99% outlier) and R00255.1 (48%). These two determinations were downweighted in the model by this proportion in each run. The former is modelled as being in the same `Phase` as OxA-X-3211-32 and appears to post-date the age of R00257.1 which is found above it. We consider the most likely explanation to be a taphonomic or contextual one, since the radiocarbon determinations appear to be accurate based on the analytical and collagen yield data shown in Table 7.

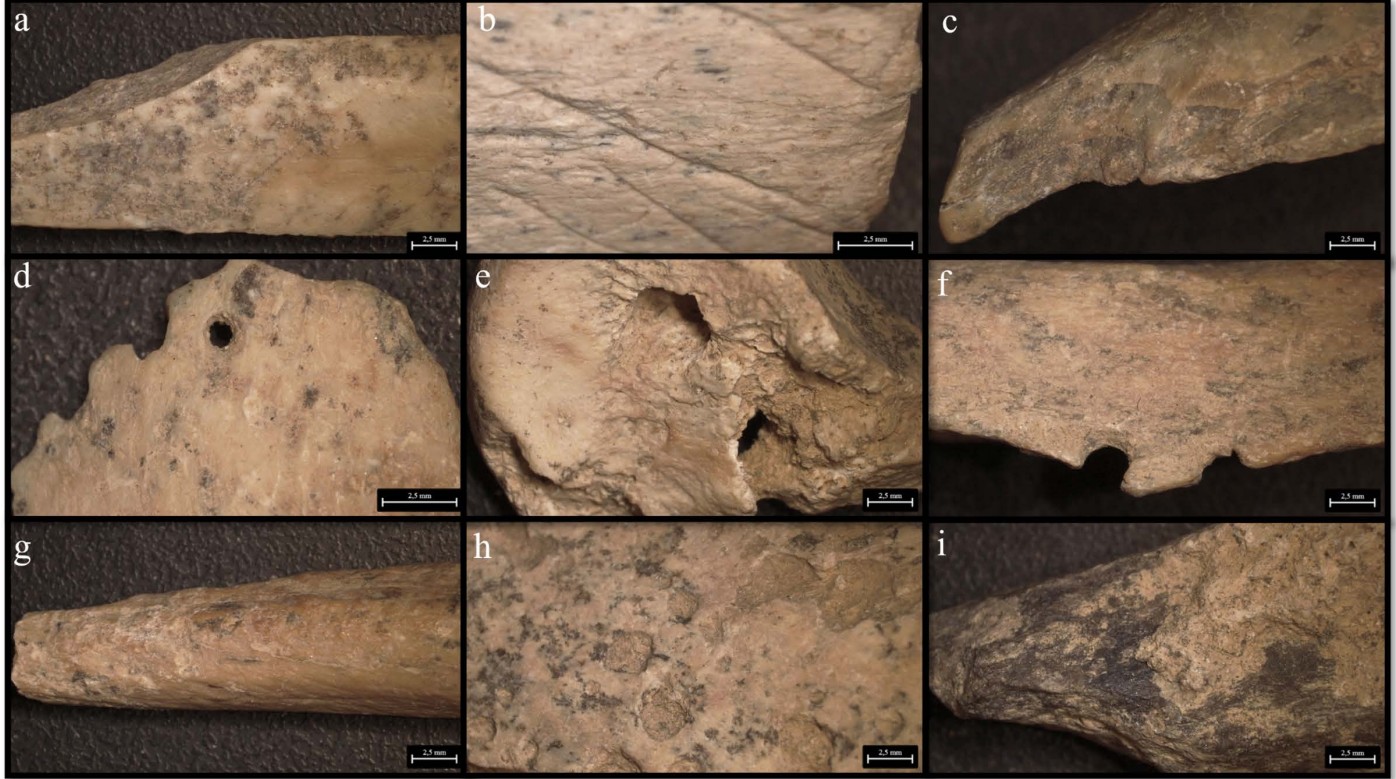

**Fig 20. Buca della lena bone's surfaces alterations.** a) Cut mark from C7; b) Cut mark from F15; c) Butchery mark form D9; d) Digested fragment by hyena from F15; e-f) Punctures and gnaw marks by hyena from F15; g) Nibbling stick from F15; h) Ichnotraces by saprophagous insects from F15; i) Manganese patina from F15.

**Table 6. NISP and MNI of Grotta del Capriolo.**

|  | A2 | | | | A3 | | | | Total | | | |
|---|---|---|---|---|---|---|---|---|---|---|---|---|
|  | NISP | % | MNI | % | NISP | % | MNI | % | NISP | % | MNI | % |
| *Felis silvestris* | 0 | 0,0% | 0 | 0,0% | 1 | 16,7% | 1 | 25,0% | 1 | 9,1% | 1 | 12,5% |
| *Ursus sp.* | 1 | 20,0% | 1 | 25,0% | 2 | 33,3% | 1 | 25,0% | 3 | 27,3% | 2 | 25,0% |
| *Rhinocerotidae* | 1 | 20,0% | 1 | 25,0% | 0 | 0,0% | 0 | 0,0% | 1 | 9,1% | 1 | 12,5% |
| *Equus ferus* | 0 | 0,0% | 0 | 0,0% | 1 | 16,7% | 1 | 25,0% | 1 | 9,1% | 1 | 12,5% |
| *Sus scrofa* | 1 | 20,0% | 1 | 25,0% | 0 | 0,0% | 0 | 0,0% | 1 | 9,1% | 1 | 12,5% |
| *Capreolus capreolus* | 2 | 40,0% | 1 | 25,0% | 2 | 33,3% | 1 | 25,0% | 4 | 36,4% | 2 | 25,0% |
| Identified specimens | 5 | 4,3% |  |  | 6 | 3,8% |  |  | 11 | 4,0% |  |  |
| Indeterminate bones | 112 | 95,7% |  |  | 150 | 96,2% |  |  | 262 | 96,0% |  |  |
| Total | 117 | 100% |  |  | 156 | 100% |  |  | 273 | 100% |  |  |

The end boundary in the model ranged between 42500—39520 cal BP, with an asymmetric distribution towards the older end of the range due to a lack of constrain to the younger end, suggesting this is, at 95.4% probability, the likely end date for the Mousterian occupation of the site. This age range is consistent with the ranges identified across Europe for the final Mousterian [9].

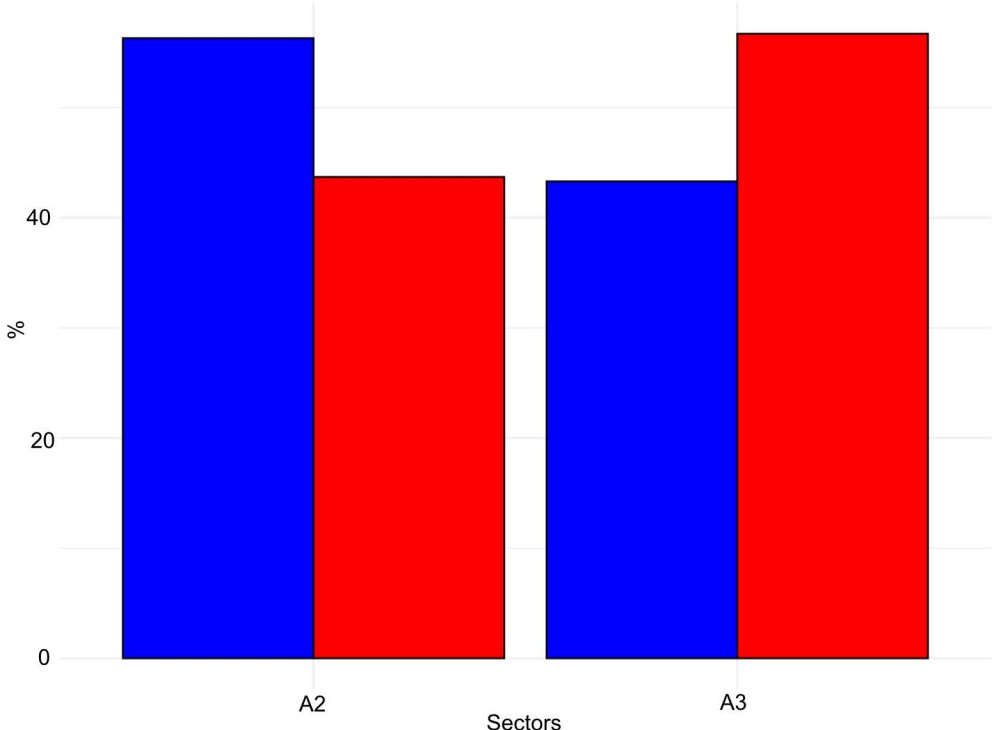

**Fig 21. Grotta del Capriolo frequency of bones with taphonomical traces (red) and without taphonomical traces (blue).** Data is expressed in percentage and organised according to excavation sectors.

## Discussion/conclusion

In this section, we evaluate the results of our analyses to assess the stratigraphical integrity of Buca della Iena and Grotta del Capriolo. Additionally, we contextualise these findings within the broader evidence for the Middle to Upper Palaeolithic transition in stratified sites of the northwestern Italian Peninsula.

### Stratigraphical integrity

The two sites were excavated with methods that do not reflect modern standards. They were excavated completely, so no material is left for new analysis. An assessment of stratigraphical integrity is, therefore, only possible through the combination of fragmentary fieldwork notes [59], comparison with the current state of the sites, and renewed analyses of the excavated findings.

The **Buca della Iena** deposit can be divided into two areas: sectors C, D, and F, and sectors E, A, and B. The latter group is likely to be severely affected by post-depositional reworking. In fact, the sequences in these sectors are shorter in extent and lack stratigraphical resolution. In sectors A and B, most lithic artefacts come from the humic layer or the highest lithological layer (spits 0 and 1). Sector E corresponds to the back of the cave, as indicated by the artefacts, which are only reported as coming from spit X, a combination of spits 8, 9, and 10 from elsewhere. It is conceivable that this area shows less stratigraphical resolution than others, so the excavators' correlation with similarly numbered spits should be approached with caution. For instance, the radiocarbon date we obtained from the sector E Level 6 (R00255.1/ UCIAMS-286278) shows a wide age range, with the lower end predating the start of human occupation.

Sectors C, D, and F are the most reliable for reconstructing the sequence, as sectors C and F form the final profile on which the stratigraphical analysis is based. The upper part of the sequence has been disturbed by agricultural activities;

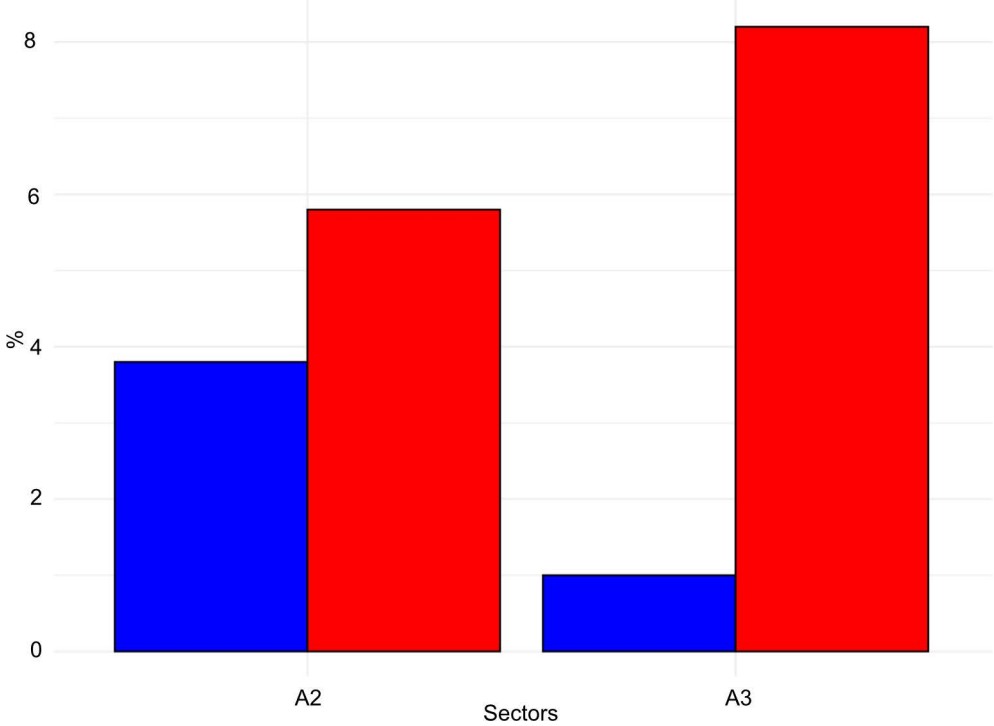

**Fig 22. Grotta del Capriolo frequency of human (blue) and small carnivores (red) identified alterations on bones.** Data is expressed in percentage and organised according to excavation sectors.

however, only a few artefacts have been found in these Levels. The bulk of artefacts are found from Level 5 onwards. The profile does not show any significant sediment sloping or discontinuities within lithological layer B. Dates from Levels 5–14 (R00254.1/UCIAMS-286277, R00253.1/UCIAMS-286276, R00259.2/VERA-8488, R00257.1/UCIAMS-286280) are consistent in indicating gradual deposit aggradation.

Sector F does show some sloping, as it forms the frontal end of the deposit. In this area, the flowstone is thinner, and the original excavator (G. Fornaciari) noted patchiness. Beneath the flowstone, no lithic artefacts have been found. The presence of coprolites and neonatal or juvenile cave hyena individuals suggests that at least this Level represents a cave hyena accumulation. Further support for this interpretation is provided by the taphonomic condition of certain anatomical elements, which exhibit damage patterns indicative of systematic bone exploitation by hyenas [84]. The collection and gnawing of shed fallen cervid antlers by cave hyenas during the Late Pleistocene is well documented [108,109] and shares similarities with those identified in other cave hyena dens [70]. The substantial number of *C. elaphus* and *M. giganteus* antler fragments found at the site does not necessarily indicate a preferential hunting choice by the predators. Instead, it reflects the scavenging behaviour typical of cave hyenas, which are known to accumulate naturally shed antlers [108,109].

Finally, partially digested long bone fragments and "nibbling sticks" were recovered. The regurgitation of long bone fragments is typical of a few middle- and large-sized carnivores (e.g., canids, cave hyenas, cave lions) during the Late Pleistocene, but cave hyena is the most represented in Buca della Iena [110].

Therefore, the most parsimonious explanation for the radiocarbon date of the bone with cutmarks in sector F spit 15 (R00256.2/VERA-8487) and the presence of other bones with cutmarks in the same sector, but the absence of lithic artefacts, is the limited runoff of archaeological material from the upper layers, accumulating at the end of the slope.

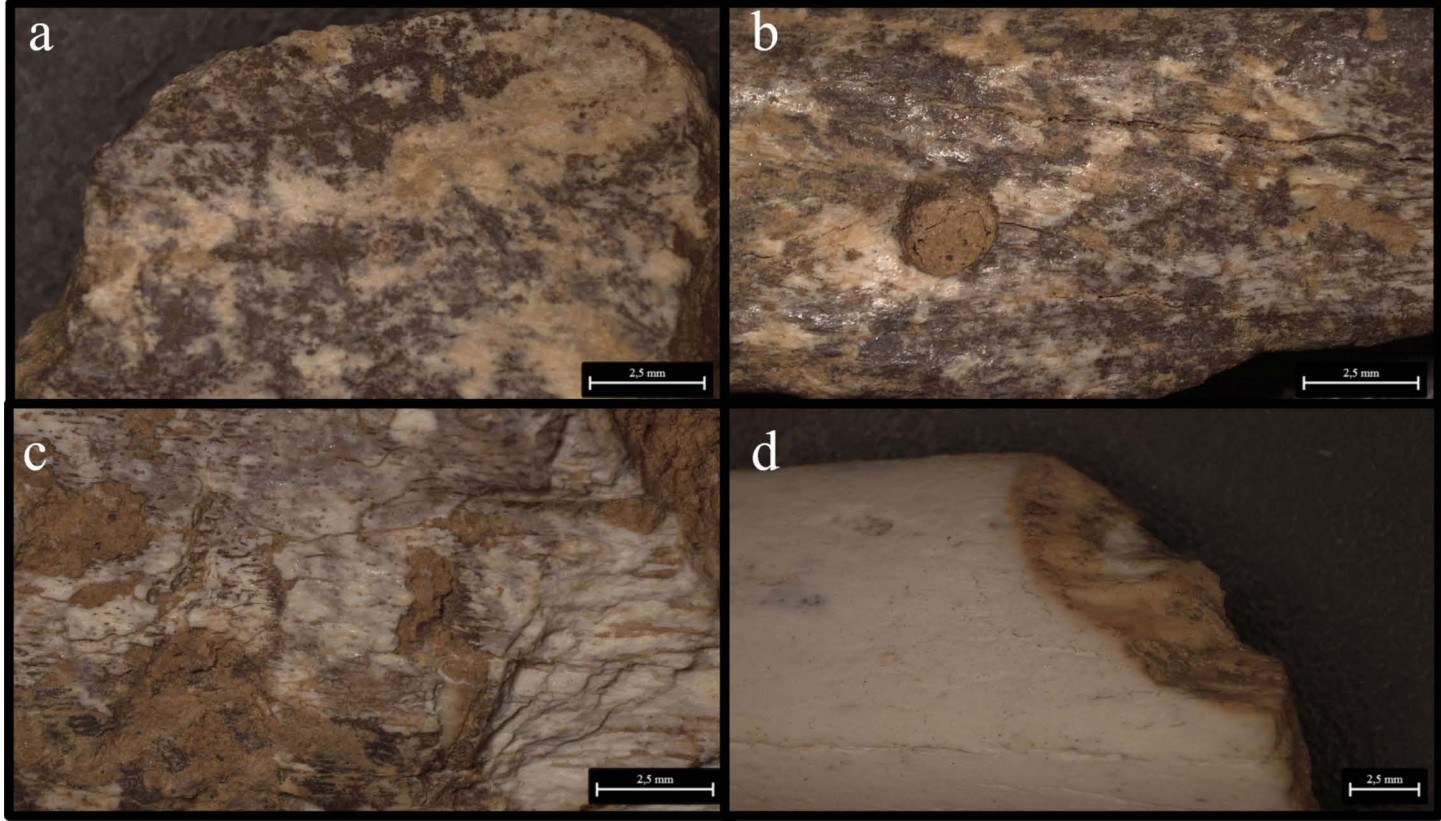

**Fig 23. Grotta del Capriolo bone's surfaces alterations.** a) Cut mark and manganese patina from A2-4; b-c) Gnaw marks and manganese patina from A3-4; d) Gnaw marks from A2-4.

**Grotta del Capriolo** deposit is more difficult to interpret, as only a few pieces of fieldwork documentation have survived. As mentioned above, based on the photos and profile drawings, it seems that deposit aggradation occurred in the centre of the site, with sediment gradually sloping due to gravity towards the valley, the back of the karst cave, and the sides. Hence, only the central part of the deposit may have preserved stratigraphical integrity. However, dating is unhelpful in assessing this aspect, as the only radiocarbon date from sector A2 (R00266.1/UCIAMS-286282) spans a wide range, extending beyond the radiocarbon limit and covering the early part of the deposition chronology. Still, the bone was recovered from the central part of the deposit (Level 4). The other radiocarbon dates (R00262.2/VERA-8489, R00264.1/UCIAMS-286281, R00271.1/UCIAMS-286509) do not show any clear stratigraphical sequence. The zooarchaeological study—both taxonomic and taphonomic—does not provide additional insights or hypotheses. The findings suggest limited human interaction with the faunal remains and minimal post-depositional modification by small carnivores and environmental conditions.

Therefore, Grotta del Capriolo's stratigraphical integrity may be largely compromised due to unrecognised post-depositional processes or mixing that occurred during deposition.

### Grotta del Capriolo and Buca della Iena in the early and mid MIS 3 northwestern Italian coast landscape

The effort to establish technocomplexes to improve the understanding of human networks and relationships is a cornerstone of archaeology. Findings from Buca della Iena and Grotta del Capriolo are typically compared with nearby sites such

**Table 7. Radiocarbon and stable isotope results obtained from 6 samples from Buca della Iena and 4 samples from Grotta del Capriolo.**

| Lab sample number | Sample reference | %N Bone | Start weight (mg) | Yield (mg) | Collagen yield (%) | P Code | F14C | F14C ±error | ¹⁴C Age | ¹⁴C Age ±error | δ15N (‰) | δ13C (‰) | C:N ratio |
|---|---|---|---|---|---|---|---|---|---|---|---|---|---|
| R00253.1/UCIAMS-286276 | BdI-2 | 1.18 | 515 | 31.08 | 6 | UF | 0.0098 | 0.0007 | 37190 | 600 | 4.46 | -20.3 | 3.13 |
| R00254.1/UCIAMS-286277 | BdI-3 | 2.06 | 551 | 27.25 | 4.9 | UF | 0.0086 | 0.0007 | 38240 | 690 | 12.14 | -19.26 | 3.12 |
| R00255.1/UCIAMS-286278 | BdI-4 | 0.73 | 568 | 8.92 | 1.6 | UF | 0.0037 | 0.0007 | 44980 | 1590 | 4.38 | -21.98 | 3.12 |
| R00256.1/UCIAMS-286279 | BdI-5 | 0.85 | 564 | 2.95 | 0.5 | UF | 0.0112 | 0.0007 | 36090 | 530 | 4.97 | -20.11 | 3.13 |
| R00256.2/VERA-8487 | BdI-5 | | 614 | 8.63 | 1.4 | UF | 0.00677 | 0.00061 | 40131 | 763 | 5.28 | -19.8 | 3.3 |
| R00257.1/UCIAMS-286280 | BdI-6 | 1.01 | 543 | 11.89 | 2.2 | UF | 0.0051 | 0.0007 | 42400 | 1200 | 10.78 | -18.56 | 3.16 |
| R00259.1/UCIAMS-286507 | BdI-8 | 0.8 | 498 | 0.88 | 0.2 | ABA | 0.0094 | 0.0011 | 37460 | 980 | | | |
| R00259.2/VERA-8488 | BdI-8 | | 753 | 17.46 | 2.3 | ABA | 0.00531 | 0.00061 | 42081 | 984 | 2.92 | -21.18 | 3.14 |
| R00262.1/UCIAMS-286508 | GCa-2 | 0.25 | 517 | 2.05 | 0.4 | ABA | 0.0121 | 0.0007 | 35450 | 490 | 3.14 | -21.9 | 3.11 |
| R00262.2/VERA-8489 | GCa-2 | | 1421 | 10.79 | 0.8 | ABA | 0.00842 | 0.00063 | 38375 | 621 | 4.49 | -22.02 | 3.34 |
| R00264.1/UCIAMS-286281 | GCa-4 | 0.32 | 597 | 21.53 | 3.6 | ABA | 0.004 | 0.0007 | 44400 | 1500 | 1.81 | -21.91 | 3.08 |
| R00266.1/UCIAMS-286282 | GCa-6 | 0.53 | 788 | 14.92 | 1.9 | ABA | 0.0024 | 0.0007 | 48310 | 2400 | 2.31 | -21.11 | 3.05 |
| R00271.1/UCIAMS-286509 | GCa-11 | 0.31 | 733 | 1.91 | 0.3 | ABA | 0.0062 | 0.001 | 40900 | 1400 | | | |
| R00271.2 | GCa-11 | 0.31 | 709 | 1.99 | 0.3 | ABA | | | | | | | |

Note: 'Collagen yield' is the percent yield of extracted collagen as a function of the starting weight of the bone analysed. The 'P code' (pretreatment code) indicates whether samples were subjected to ultrafiltration (UF) or not (ABA). The 'C:N ratio' is the carbon to nitrogen atomic ratio. It should range between 2.9–3.5. The theoretical value is 3.18. Samples highlighted in grey produced low collagen yields and ¹⁴C ages that are too young. Samples below these marked with a '.2' subsample number represent second collagen extractions done on the same material. In three of the four cases this results in increased collagen yields and ¹⁴C ages.

as Tecchia di Equi, Grotta all'Onda, and Buca del Tasso. However, these three sites present some challenges, primarily due to outdated excavation methods.

Tecchia di Equi has been investigated intermittently since 1911 [46]. Despite a rich record of Pleistocene fauna and lithic materials, the stratigraphical context of much of the collection is poorly understood or lost. The first investigator, De Stefani, discovered the main karst tunnel and reported two main layers of occupation [46,111], but the excavation methods were too crude to properly assess post-depositional mixing. In 1933, attention shifted to the exterior shelter area [112], where two occupation layers were found, although the artefacts from these layers have since been lost. Later investigations (1967, 1970–1974, 1980, 1982, 1997) unearthed disturbed deposits or explored areas that were too small to clarify the site's stratigraphical relationships [113–116]. New excavations (2009–2012) provided improved stratigraphical information for the exterior shelter area and inner cave, radiocarbon dating the primary Pleistocene occupation to the mid-part of MIS 3 (43.7±1.9 ka uncal BP and 44.0±2.2 ka uncal BP) during a cold climatic spell [47,117,118]. The lithic materials from these excavations remain largely unstudied, aside from a preliminary typological assessment of De Stefani's findings(53) and brief techno-typological observations [47,48]. Preliminary analyses (J.G.) indicate a predominance of Levallois centripetal recurrent methods.

Grotta all'Onda was mostly investigated before World War II (1914, 1931), with more recent campaigns conducted between 1996 and 2005 [50,119–122]. Both the old and new excavations focused on the cave entrance, progressively extending closer to the western cave wall [120,122]. Stratigraphical assessments indicate a sequence of clay and silty layers between two main flowstones [120]. The upper flowstone marks the Pleistocene-Holocene transition, while the lower one is dated to 174±8.2 ka BP [51]. A 1968 date from a flowstone sample taken in 1931 (39.3±3.2 ka BP) led to speculation about intermediate flowstone levels that are now lost [51,57]. The 1968 date was considered a maximum for the Pleistocene deposit above, containing Mousterian artefacts (Layer C). New radiocarbon dates on bones recovered during recent excavations in a context similar to Layer C yielded radiocarbon ages of 37.1±0.5 ka uncal BP (42.5–40.7 ka cal

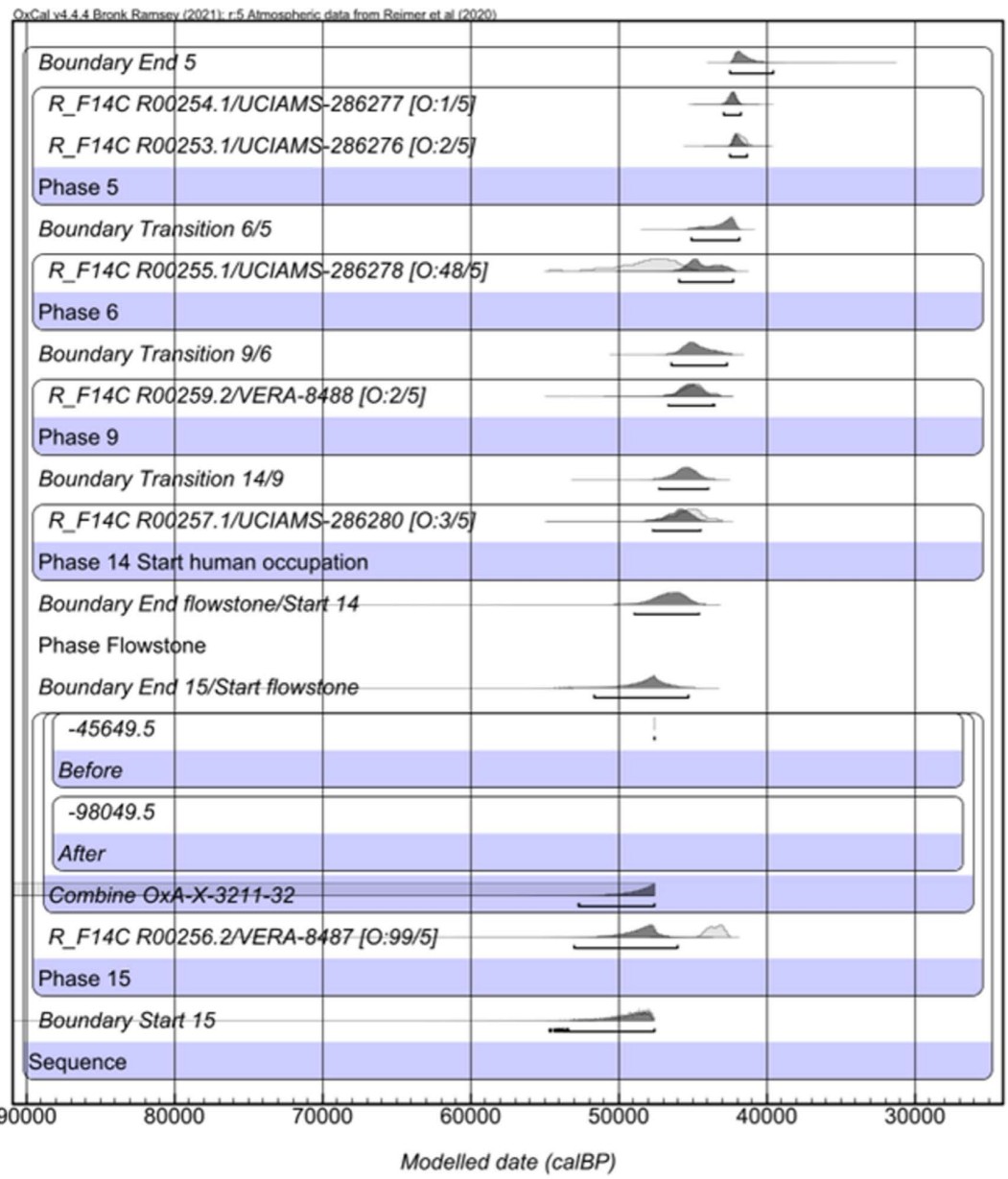

**Fig 24. Bayesian age modern for Buca della lena.** Built using OxCal 4.4 [106] and the INTCAL20 curve [107].

BP) and 37.0±0.6 ka uncal BP (42.4–40.5 ka cal BP) [21]. However, the small lithic assemblage and unclear relationship with the cave bear (U. spelaeus), who dominates the faunal assemblage, complicate interpretations [123]. As with Tecchia di Equi, the lithic assemblage has not been fully analysed, and only brief mentions of findings exist [48].

Buca del Tasso was completely excavated in 1919–1922 [52]. The deposit featured three clay horizons (Layer C, levels III, II, I), a breccia deposit (Layer B), and a top clay layer (Layer A) with a darker lens, interpreted as a human occupation layer. Recently, radiocarbon dates on bones from Layer A provided a radiocarbon age of 40.2±1.3 ka BP (45.9–42.2 ka cal BP), while Layer C was dated to 48.1±3.4 ka BP (>46.6 ka cal BP). A caprine bone from Layer A was radiocarbon

dated to 333 ± 17 ka BP, suggesting some mixing in the upper layer. The lithic assemblage is small, with only seven artefacts from Layer C level III and 57 from Layer A [52]. Techno-typological analysis suggests a generic Mousterian attribution with the Levallois method, with Layer A showing more elongated products, similar to those in the lower levels of Grotta del Capriolo [48].

Given the proximity of these three sites and their early excavations, findings from Grotta del Capriolo and Buca della Iena have been compared to them. All the sites were originally associated with denticulate tools and placed within the same technocomplex, the Alpine Mousterian, representing Neanderthal adaptation to higher-elevation environments [48,56]. However, new research suggests that many of the denticulates from Buca della Iena and Grotta del Capriolo are actually the result of taphonomic damage [124]. The new radiocarbon dates from Buca della Iena and Grotta del Capriolo suggest that northwestern Tuscany was home to Neanderthal groups using Levallois technology during mid-MIS 3.

Expanding the geographical scope, other nearby dated Mousterian sites are located in southern Tuscany and western Liguria. In western Liguria, the Balzi Rossi complex features long depositional sequences that include late MIS 3 Mousterian layers [125]. Riparo Mochi and Riparo Bombrini show a similar chrono-cultural pattern [22,23]. Riparo Mochi Unit I contains the latest Mousterian occupations with an end boundary ranging between 45.2–41.7 ka cal BP at 95.4% probability), which has been interpreted as likely occurring during a dry and cold climatic spell [15,23]. The deposit was excavated in arbitrary 5–10 cm spits, and the upper part of the Mousterian unit consists of spits 56–31 [15]. A technological change occurs around spit 56, as elongated Levallois products become rarer, while shorter Discoid products become more abundant [15]. A new interpretation shows that the two technologies might be part of different phases of the same reduction sequence: the decrease in elongated products is mostly due to the smaller number of regional lithotypes [15]. Riparo Bombrini features a shorter sequence than Mochi, consisting only of mid-MIS 3 deposits, the Mousterian sits within the lowest units labelled MS or M and with sequential numbers [22]. The upper levels (MS) feature a low density of finds (the S stands for "sterile") [126]. M4 is dated to the 46–44 ka cal BP interval (45.8–44.4 ka cal BP at 68.2% probability, 46.2–44.3 ka cal BP at 95.4% probability) and the earliest Aurignacian is dated to the 43–42 ka cal BP interval (43.4–42.0 ka cal BP at 95.4% probability): therefore the Riparo Bombrini Mousterian is assumed to cover this short span (22,126). As for lithic technology, results are similar to Riparo Mochi: Discoid products are most common and they are usually made on local raw materials, while Levallois implements are rare [22]. The MS levels might identify an ephemeral use of the area by the last Neanderthals exploiting a smaller geographical range [126]. More to the east, MIS 3 Mousterian archaeological layers are found in Arma delle Manie and Santa Lucia Superiore [126,127]. The Arma delle Manie lowest archaeological unit (VII) is dated to 60 ± 9 ka BP, units IV to II to an interval between 39 and 33 ± 5 ka cal BP [13,127]. Discoid and other shorter flakes technologies characterise the assemblages from VII to I, Levallois presence is constant but rarer [127]. No chronological determinations are available for Santa Lucia Superiore, but pollen analysis shows Layer B is mostly related to MIS 3 [127]. The assemblage is similar to the other Ligurians MIS 3 Mousterian assemblages, with a higher presence of Discoid and few Levallois products [127].

In southern Tuscany, Grotta La Fabbrica shows one of the most important stratigraphical sequences in Central Italy for MIS 3 as it preserves the whole extent of Middle-to-Upper Palaeolithic Transition [27,128,129]. The sequence consists of the basal Layer 1, containing Mousterian industry, Layer 2, containing Uluzzian industry, Layers 3–4, containing Aurignacian industry, and Layer 5, containing Gravettian or Epigravettian industry [27]. The cave has an inner slope and most of the deposit has been reworked by post-depositional processes, though a small area next to the northern area of the cave is *in situ*, the transition between the layers shows erosive unconformities [130]. Layer 1 is divided into several sub-levels, 1a being the uppermost and dated to 44.0 ± 2.2 ka BP with Optically Stimulated Luminescence (OSL) dating [27]. The assemblage is characterised by Levallois cores, mostly recurrent with centripetal modality, but Levallois flakes are rare [27,130]. Another recently re-excavated site is Cala dei Santi. The site consists of fine sandy and clay layers interstratified by several thin flowstones [21]. The main archaeological occupations occur in units 20.2-1004 and 20.3; sub-units 20.2.4-1004A, 20.2.2-1004B, 20.3.1-150, 20.3.2-111, and 20.3.2-110 preserves living floors with hearths, accumulation of ash

and organic matter and lithics. Using radiocarbon and OSL dating, the lowermost unit 20.2-1004 starts around 50.4–46.9 ka cal BP and ends around 48.7–45.5 ka cal BP (95.4% probability). The uppermost sub-unit 20.3.2-110 starts around 47.0–40.3 ka cal BP and ends around 46.3–38.1 ka cal BP (95.4% probability) [21]. The large timespan, place the upper sub-unit well above the likely end of the Mousterian in the Italian Peninsula; the most likely date from sub-unit 20.3.2-110 is the R-EVA 9282010 (45.6±530 ka BP) [21]. The lithic assemblages are still under analysis, ascribed to the Late Mousterian [21]. Only the 20.3.1-150A assemblage has been preliminarily studied, it consists of a small assemblage which is characterised by unidirectional recurrent Levallois and prismatic unidirectional flaking [131].

Therefore, Buca della Iena and Grotta del Capriolo, despite their limitations, offer a good chance to cover a research gap between western Liguria and southern Tuscany. The new radiocarbon dates we obtained show that the two sites were frequented pene-contemporaneously within the 50–40 ka cal BP timespan, making the sites pivotal in assessing Neanderthal behavioural and evolutionary trajectories at the end of their history. Despite the long research history in northwestern Tuscany, the Aurignacian remains elusive and needs to be found in a stratigraphical sequence comprising the Mousterian. Few artefacts have been reported from Grotta all'Onda, but they are mostly undiagnostic or coming from a reworked context. From the unstratified lithic collections of Massaciuccoli, some Aurignacian lithics are reported [132], but no updated analysis confirms it. Pontecosi [133] is the only unequivocal Upper Palaeolithic site in the wider area, but it is currently not radiometrically dated and the site context (open-air, no organic preservation, destroyed by modern construction works) does not speak favourably about the chance of dating it. Also, techno-typological characteristics might derive from a later stage of the Aurignacian. The upper levels of Buca della Iena and the younger dating of Grotta del Capriolo overlap chronologically with the last Neanderthal occupation in the Balzi Rossi sites. They also overlap behaviourally, as they show intensive use of local resources and high mobility of Neanderthal groups within these small territories, as evidenced by the small assemblages and repeated visits to the same sites [15,126]. This pattern might be shown by the small assemblages so far found in the northwestern Tuscany sites, which do not mirror long-term occupations of the sites. Occupations in Buca della Iena and Grotta del Capriolo are broadly contemporaneous and therefore highlight a consistent use of the area by Neanderthals throughout the 50–40 ka cal BP period. This likely involved negotiating a relationship with the nearby Buca della Iena hyaenas, either avoiding or engaging them. The small percentage of anthropogenic modifications found on Buca della Iena fauna remains and the small lithic assemblages of each level might suggest the first option. Nevertheless, some engagement might be at play as the hyaena dated radius in Level 5 of Buca della Iena shows anthropogenic cutmarks. In addition, the technology employed in the two areas broadly matches, with a higher reliance on the Levallois method in northwestern Tuscany, which may be related to the available raw materials. In southern Tuscany, Grotta La Fabbrica and Grotta dei Santi, mostly show that Neanderthal occupations ended earlier than in the northern Tyrrhenian-Ligurian coast. Especially, in Grotta La Fabbrica, the Uluzzian date shows that *H. sapiens* groups replaced Neanderthals at the time of the last Neanderthal occupations in Riparo Mochi, Bombrini, Buca della Iena, and Grotta del Capriolo.

We have presented a comprehensive re-evaluation of two archaeological sites that have been often related to the Middle-to-Upper Palaeolithic Transition in northwestern Tuscany: Buca della Iena and Grotta del Capriolo. Research at the sites has been hampered by the small extent of the archaeological deposits, the outdated recovery strategies and the fact that all the cultural material was excavated. Nevertheless, we show that re-assessment of these contexts can shed new light on important archaeological questions. The combination of fieldwork documentation re-assessment, technological lithic analyses, faunal taphonomical analyses, and radiometric dating has shed some new light on previously unknown site histories. Buca della Iena and Grotta del Capriolo witnessed repeated late Neanderthal occupations, and their re-evaluation is important in the understanding of Neanderthal extinction and local Transition to *H. sapiens* dynamics.

## Supporting information

**S1 File.  Oxcal code and Bayesian model results generated for Buca della Iena.**
(DOCX)

**S2 File. Buca della Iena and Grotta del Capriolo cores technological dataset.**
(XLSX)

**S3 File. Buca della Iena and Grotta del Capriolo debitage technological dataset.**
(XLSX)

**S4 File. Buca della Iena frequency of taphonomical alterations.**
(XLSX)

**S5 File. Buca della Iena frequency and origin of taphonomical alterations.**
(XLSX)

**S6 File. Grotta del Capriolo frequency of taphonomical alterations.**
(XLSX)

**S7 File. Grotta del Capriolo frequency and origin of taphonomical alterations.**
(XLSX)

**S8 File. Buca della Iena spits correlation across trenches and attribution to lithological layer.**
(XLSX)

**S9 File. Grotta del Capriolo spits correlation across trenches and attribution to lithological layer and new levels.**
(XLSX)

**S10 File. Grotta del Capriolo and Buca della Iena total numbers of lithic artefacts and faunal remains divided by stratigraphical levels within the stratigraphically reliable areas.**
(XLSX)

## Acknowledgments

We are grateful to the individuals and institutions that supported and allowed the investigation. Jacopo Gennai is currently supported by the Progetto di Eccellenza 2023–2027 "Un senso nel disordine. Praticare la complessità" awarded to the Civilisations and Forms of Knowledge department. Prof. Gino Fornaciari and Prof. Carlo Tozzi for providing the existing documentation in their possession. SABAP Lucca e Massa e Carrara authorised the new investigations on the Buca della Iena, and Grotta del Capriolo archaeological collections. We are grateful to the staff of the Museo Civico Archeologico e dell'Uomo "Carlo Alberto Blanc" in Viareggio for their assistance during the research stays of Jacopo Gennai, Marco Romboni, and Angelica Fiorillo. Mr. Gabriele Orlandi provided access to his property to survey Buca della Iena and allowed taking pictures. We thank the two anonymous reviewers for suggesting improvements to the manuscript.

## Author contributions

**Conceptualization:** Jacopo Gennai, Marco Romboni.

**Data curation:** Jacopo Gennai, Tom Higham, Marco Romboni, Angelica Fiorillo, Laura van der Sluis.

**Formal analysis:** Jacopo Gennai, Tom Higham, Angelica Fiorillo, Maddalena Giannì, Laura van der Sluis.

**Funding acquisition:** Jacopo Gennai, Damiano Marchi, Elisabetta Starnini.

**Investigation:** Jacopo Gennai, Tom Higham, Angelica Fiorillo, Maddalena Giannì.

**Methodology:** Jacopo Gennai, Tom Higham, Marco Romboni, Angelica Fiorillo.

**Project administration:** Jacopo Gennai.

**Resources:** Jacopo Gennai, Tom Higham, Damiano Marchi, Elisabetta Starnini.

**Supervision:** Tom Higham, Damiano Marchi, Elisabetta Starnini.

**Validation:** Tom Higham.

**Visualization:** Jacopo Gennai, Tom Higham.

**Writing – original draft:** Jacopo Gennai, Tom Higham, Marco Romboni, Angelica Fiorillo, Maddalena Giannì, Laura van der Sluis, Damiano Marchi, Elisabetta Starnini.

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
