## [Decision Letter · Decision Letter 0]

4 Feb 2025

PONE-D-24-55044Buca della Iena and Grotta del Capriolo: new chronological, lithic, and faunal analyses of two late Mousterian sites in Central ItalyPLOS ONE

Dear Dr. Gennai,

Thank you for submitting your manuscript to PLOS ONE. After careful consideration, we feel that it has merit but does not fully meet PLOS ONE’s publication criteria as it currently stands. Therefore, we invite you to submit a revised version of the manuscript that addresses the points raised during the review process.

We look forward to receiving your revised manuscript.

Kind regards,

Enza Elena Spinapolice, Ph.D

Academic Editor

PLOS ONE

3. In your manuscript, please provide additional information regarding the specimens used in your study. Ensure that you have reported human remain specimen numbers and complete repository information, including museum name and geographic location.

For more information on PLOS ONE's requirements for paleontology and archeology research, see https://journals.plos.org/plosone/s/submission-guidelines#loc-paleontology-and-archaeology-research.

4. In your Methods section, please provide additional information regarding the permits you obtained for the work. Please ensure you have included the full name of the authority that approved the field site access and, if no permits were required, a brief statement explaining why.

“The research is funded by the Horizon Europe scheme (GA no. 101061427 Acronym: MobiliTy) awarded to Jacopo Gennai and Elisabetta Starnini. Dates of Buca del Tasso and one date of Buca della Iena was funded by the Center Museo di Storia Naturale of the University of Pisa awarded to Damiano Marchi.”

6. We note that Figures 1, 2, 4, and 18 in your submission contain [map/satellite] images which may be copyrighted. All PLOS content is published under the Creative Commons Attribution License (CC BY 4.0), which means that the manuscript, images, and Supporting Information files will be freely available online, and any third party is permitted to access, download, copy, distribute, and use these materials in any way, even commercially, with proper attribution. For these reasons, we cannot publish previously copyrighted maps or satellite images created using proprietary data, such as Google software (Google Maps, Street View, and Earth). For more information, see our copyright guidelines: http://journals.plos.org/plosone/s/licenses-and-copyright.

1. You may seek permission from the original copyright holder of Figures 1, 2, 4, and 18 to publish the content specifically under the CC BY 4.0 license. 

7. We note that Figures 6, 7, 8, 12, 13, and 14in your submission contain copyrighted images. All PLOS content is published under the Creative Commons Attribution License (CC BY 4.0), which means that the manuscript, images, and Supporting Information files will be freely available online, and any third party is permitted to access, download, copy, distribute, and use these materials in any way, even commercially, with proper attribution. For more information, see our copyright guidelines: http://journals.plos.org/plosone/s/licenses-and-copyright.

1. You may seek permission from the original copyright holder of Figures 6, 7, 8, 12, 13, and 14 to publish the content specifically under the CC BY 4.0 license.

Reviewers' comments:

Reviewer's Responses to Questions

**Comments to the Author**

1. Is the manuscript technically sound, and do the data support the conclusions?

Reviewer #1: Yes

Reviewer #2: Yes

2. Has the statistical analysis been performed appropriately and rigorously? 

Reviewer #1: N/A

Reviewer #2: Yes

3. Have the authors made all data underlying the findings in their manuscript fully available?

Reviewer #1: Yes

Reviewer #2: Yes

4. Is the manuscript presented in an intelligible fashion and written in standard English?

Reviewer #1: Yes

Reviewer #2: Yes

5. Review Comments to the Author

Reviewer #1: The paper by Gennai et al, deals with the reappraisal of the studies on the materials from two Late Mousterian sites, Buca della Iena e Grotta del Capriolo, located in Tuscany and excavated mainly in the ‘70s.

Although the paper has the merit of reanalyzing with a more modern approach two otherwise neglected sites, putting them in the context of coeval sites in NW Italy, and this is of course always welcome, possibly its contribution to “enhance our understanding of late Neanderthal settlement in the northwestern Italian peninsula and provide insights into their demise” (lines 21-22 in the abstract) is not so significant because on one side (Buca della Iena) we are dealing mainly with a hyaena den with just some evidence of Neanderthal occupation and on the other (Grotta del Capriolo) we only have a very small sample that in the end, with all its limitations, is not so informative; therefore maybe the authors should not explicitly aim at such high goals, that cannot be achieved with the information from these sites.

At the beginning of the Abstract maybe some indication at least to the Region where the sites are located should be added.

On line 22 the reference about “NW Italian peninsula” (similar indication also on line 32) can be understood only after reading the paper, but here, just reading the abstract, it is a bit confusing since the title is mentioning only “central Italy”. Try to modify the phrase, adding for example something about the fact that the sites will be considered within the context of NW Italian sites.

Line 38 “..the end of their kind…”, maybe better “ …their disappearance…” (it applies to both Neanderthals and Mousterian industries).

Lines 50-56. The Uluzzian, placed here between the two Homo sapiens dispersals, needs some connection with previous and subsequent phrases, explaining why it is mentioned there, besides simple chronology.

Line 61. “Raw material procurement in these contexts …”, although there are the references, please add information about the name and/ or location of such contexts.

Line 91 Maybe better “..new radiocarbon dates” instead of “new radiocarbon data”.

Line 144 Lithic Analysis. The fauna paragraph (see later) explains that only the sample from reliable excavation areas has been considered. Here there should be a similar explicit statement saying that in this case materials from both reliable and unreliable excavation areas will be considered and compared.

From Line 153 Fauna Analysis. As far as the faunal sample from Buca della Iena is concerned, there is absolutely no mention about the previous restudy of the assemblage by Stiner, published in her 1994 book, but considered also in others of her papers as an example of a hyaena den, therefore also these references should be taken into account and discussed explicitly also comparing information.

Besides the total number of specimens involved also the size of the two samples should be mentioned, especially because sample sizes are very different.

Info about methodologies used for assessing age at death and age categories should be added.

Lines 165-166 cut marks are one type of butchery marks, so maybe just use the latter (similar occurrences are present also in other parts of the paper, so check them too).

Line 168 it is “… gnawing by rodents and small carnivores…” and not “…of rodents…”

Table 2 and Table 3. In both tables there are several lines with numbers but no categories. Maybe I am missing something, but in any case, the meaning of such numbers is not clear. Totals of the categories (e.g., Blade, Core) should be at least plural (or explicitly “Total Blades, Total Cores) and maybe placed at the bottom and not at the top of the corresponding list. Check numbers in the tables, for example in Table 2 the stratigraphically unreliable “Blade” (total) should be 8 and not 7, and actually the grand total in the last column is correctly 11 and not 10. For better clarity please add blade/flake/core/etc. wherever only the adjective is used.

Figures 6-8 and 12-14, for an easier reading, maybe add to the captions the reference to artifact categories at least by group, not leaving all the details to the supplementary materials.

The fact that at both sites “Complete artefacts are the most frequent” is related to “modern” selection during excavation and previous studies, or could it be ancient? Please discuss and comment on this. Is debris completely missing or was it not considered in this study.

Lines 235-236. The reference to positive and negative peaks is not correct since there are no values below 0 in the graph of figure 11. Therefore, it is better to use “high” and “low”. (same in line 280).

In order to assess artifact density how was the cubic meter calculated?

In the captions of Figs 9-10 and 15-16. Better phrase “The frequency is reported as percentage using only stratigraphically reliable artefacts”.

For both sites it would be better to add a summary table with the total number of artifacts by level in order to understand better the statistical significance of the percentages reported in the graphs.

In figures 11 and 17 the term levels instead of spits that is employed in all the other lithic graphs is used, while the fauna graphs for Buca della Iena use Level ….please unify. Mention somewhere in the first part that no lithics were found in layer 15 (it only appears much later in the paper).

Line 265 Assessing the homogeneity of an assemblage by comparing two samples of very different sizes (99 vs. 440) maybe is not so statistically valid.

Line 266 Table 5 is probably Table 4.

Line 321 Please add some comments and possibly a graph with the distribution of the faunal remains by layer.

In the Fauna text and tables please correct Rhinoceratidae into Rhinocerotidae

Line 332 Please specify in the methods which animals (at these sites) fall in the three size categories.

In tables 4 and 6 the line “Identified specimens” repeats the same information of the line just above it (Total Identified specimens), so maybe remove it. Does the “Indeterminate bones” category include also vertebrae and ribs or not? In case it does not include them maybe move the line just above the final total.

Line 333 Since according to taxonomic identification U. spelaeus is the most frequent species in almost all the levels, discuss more extensively later in the paper why Buca della Iena should be a hyaena den as the name of the site itself implies and not a bear den.

Lines 341-343. This part about the age at death should be expanded in particular providing more detail by species, especially the most common ones, focusing on both carnivores (probably mostly of natural accumulation) and ungulates (result of carnivore and human accumulation?), and mentioning possible peculiarities for more rare taxa.

Lines 347-349 Here and in Fig. 19 it makes no sense to lump together all the modifications, it is better to divide them at least by group (e.g., human, carnivore, other natural)

Line 350 As mentioned before, please discuss more extensively somewhere why the carnivore traces observed are attributed only to hyaena and not to bear.

Line 377 Please specify in the discussion hypotheses about why the faunal assemblage is so poorly preserved at Capriolo.

There are some problems with the numbering of the Supporting information material in the main text (e.g., Iena and Capriolo SI fauna tables are reversed and mixed) and some of the references to the SI tables along the main text are missing. Please check and add.

Line 388 soil pH and manganese staining are two different things, please correct here and in the relevant SI tables.

Line 436 is missing the specific reference to the relevant Supplementary Information file.

Line 629 “smaller range” of what? Smaller geographical range? Please specify.

Lines 677-680 Although I may agree about the similarities in the use of local resources, I do not see any real evidence from the data presented in this paper about the high mobility at Buca della Iena and Grotta del Capriolo. Buca della Iena may have been just used ephemerally because most of the time it was a carnivore den, but this does not imply necessarily high mobility ( and of course the data for Capriolo are not enough to infer much about mobility). Therefore, support better your hypothesis of high mobility at these sites.

Although it is difficult to distinguish between faunal remains accumulated by carnivores and those accumulated by humans (but maybe shed antlers are likely just carnivore accumulation), is there any evidence of seasonal use of Buca della Iena, and are there any differences between layers in this aspect? Are there any available seasonality data for Grotta del Capriolo?

In the discussion it would be interesting to have a commented graph with the distribution of both lithics and faunal remains by layer, adding also information on human and carnivore damage on the bones and see how they fit (or not) in order to infer variations along the sequence in the type of occupation by humans and carnivores.

Reviewer #2: The article is undoubtedly essential for understanding the latest Neanderthal evidence during MIS 3 in an area of northern Tuscany, which serves as a bridge between the Ligurian final contexts and those of southern Tuscany, thus filling a gap in the occupation of Homo neanderthalensis in this area of the Italian Peninsula. The study of contexts discovered and documented in periods when archaeological techniques were not yet fully developed, such as the sites investigated for this article, like Buca della Iena and Grotta del Capriolo, is of fundamental importance. The technological analysis of the lithic industry and the faunal analysis have been carried out according to the classic and well-established criteria that underpin modern studies of these materials, all enriched by chronological analyses that allow these contexts to be framed within a sufficiently short time window, thus enabling the evaluation of Neanderthal occupation in the area. However, the occupation of the two sites remains unclear, as the two caves were likely shared by alternating occupations of Homo neanderthalensis, Crocuta crocuta, and Ursus spelaeus. Perhaps the dynamics of occupation at these two sites should be better explained, as has been done for Grotta Guattari at Circeo. Given the age of these contexts and the scarce documentation, I am uncertain whether such an explanation is entirely feasible in detail. The images proposed are suitable for the article; my only critique pertains to the images related to the lithic industry, which, in my opinion, could be enhanced with the addition of diacritical schematic arrows to highlight the directions of detachment on the flakes and cores.

6. PLOS authors have the option to publish the peer review history of their article (what does this mean? ). If published, this will include your full peer review and any attached files.

**Do you want your identity to be public for this peer review?** For information about this choice, including consent withdrawal, please see our Privacy Policy .

Reviewer #1: No

Reviewer #2: No

---

## [Author Response · Author response to Decision Letter 1]

19 Mar 2025

Response to Reviewers “Buca della Iena and Grotta del Capriolo: new chronological, lithic, and faunal analyses of two late Mousterian sites in Central Italy”

Dear Academic Editor and Anonymous Reviewers,

We sincerely appreciate your positive assessment of our research and recognition of its merit for publication in PLOS ONE. We are also grateful for your constructive comments, which have helped improve the manuscript.

Below, we address each of your comments individually, providing detailed responses and justifications. Reviewer comments appear in italics, followed by our responses in regular text.

Reviewer #1:

The paper by Gennai et al, deals with the reappraisal of the studies on the materials from two Late Mousterian sites, Buca della Iena e Grotta del Capriolo, located in Tuscany and excavated mainly in the ‘70s. Although the paper has the merit of reanalyzing with a more modern approach two otherwise neglected sites, putting them in the context of coeval sites in NW Italy, and this is of course always welcome, possibly its contribution to “enhance our understanding of late Neanderthal settlement in the northwestern Italian peninsula and provide insights into their demise” (lines 21-22 in the abstract) is not so significant because on one side (Buca della Iena) we are dealing mainly with a hyaena den with just some evidence of Neanderthal occupation and on the other (Grotta del Capriolo) we only have a very small sample that in the end, with all its limitations, is not so informative; therefore maybe the authors should not explicitly aim at such high goals, that cannot be achieved with the information from these sites.

We respectfully disagree with this assessment. As demonstrated throughout the manuscript, Buca della Iena and Grotta del Capriolo are among the few stratified sites with Mousterian artefacts in Tuscany and northwestern Italy. To find equivalent stratified occurrences someone needs to travel to Western Liguria or Southern Tuscany, hundreds of km away. The stratified nature is crucial for a reliable chronological assessment. Buca della Iena presents a well-stratified sequence with a clear chronological aggradation, bracketing the deposit between 50 and 40 ka cal BP, with human visits dated between 47 and 42 ka cal BP. Grotta del Capriolo also falls within the 50–40 ka cal BP range, although its dates do not show a fully reliable aggradation.

Although the lithic assemblages are small, these sites contain the richest stratified Mousterian collections in the region. Other sites, such as Grotta all’Onda and Buca del Tasso, contain only a few dozen artefacts, highlighting the significance of Buca della Iena and Grotta del Capriolo as key references for late Neanderthal occupation in this part of northwestern Italy. Their stratified deposits provide crucial evidence for pinpointing Neanderthal presence in the study area at the time of their final demise (43-40 ka cal BP).

Nevertheless, in response to the reviewer’s concern, we have replaced the term "settlement" with the more neutral "occupation" to more accurately reflect the nature of the evidence.

At the beginning of the Abstract maybe some indication at least to the Region where the sites are located should be added.

On line 22 the reference about “NW Italian peninsula” (similar indication also on line 32) can be understood only after reading the paper, but here, just reading the abstract, it is a bit confusing since the title is mentioning only “central Italy”. Try to modify the phrase, adding for example something about the fact that the sites will be considered within the context of NW Italian sites.

We provided better geographical contextualization.

Line 38 “..the end of their kind…”, maybe better “ …their disappearance…” (it applies to both Neanderthals and Mousterian industries).

We thank the reviewer for the suggestion that we have included in the manuscript text.

Lines 50-56. The Uluzzian, placed here between the two Homo sapiens dispersals, needs some connection with previous and subsequent phrases, explaining why it is mentioned there, besides simple chronology.

The whole introductive section is devoted to explain the state-of-the-art of the Middle to Upper Palaeolithic of the study area and the broader Northwestern Mediterranean area. Therefore, we find appropriate to mention the Uluzzian. We restructured the whole section so the meaning is clearer.

Line 61. “Raw material procurement in these contexts …”, although there are the references, please add information about the name and/ or location of such contexts.

We provided the names of the sites cited (Riparo Mochi and Riparo Bombrini).

Line 91 Maybe better “..new radiocarbon dates” instead of “new radiocarbon data”.

We have corrected accordingly.

Line 144 Lithic Analysis. The fauna paragraph (see later) explains that only the sample from reliable excavation areas has been considered. Here there should be a similar explicit statement saying that in this case materials from both reliable and unreliable excavation areas will be considered and compared.

We have complied with the reviewer’s suggestion and have further clarified that all lithic artefacts marked as originating from Buca della Iena and Grotta del Capriolo were analysed, regardless of their assigned stratigraphical unit or contextual information.

From Line 153 Fauna Analysis. As far as the faunal sample from Buca della Iena is concerned, there is absolutely no mention about the previous restudy of the assemblage by Stiner, published in her 1994 book, but considered also in others of her papers as an example of a hyaena den, therefore also these references should be taken into account and discussed explicitly also comparing information.

We appreciate the reviewer’s insights. However, they did not acknowledge that Mary Stiner’s work is not directly comparable to our analysis due to significant inconsistencies, particularly regarding the stratigraphical sequence in which Mary Stiner subdivided her sample. Her presentation of Buca della Iena in Honor among Thieves: A Zooarchaeological Study of Neandertal Ecology (1994, Chapter 3, p. 66) is rather vague and, in some instances, incorrect. Notably, Buca della Iena and Grotta del Capriolo were not excavated in 1964–65, as stated in her work. Instead, Buca della Iena was excavated in 1966, and Grotta del Capriolo in 1968, as we clearly documented in our manuscript with supporting field notes. While Stiner provided careful descriptions of the Latium sites, no tables or figures illustrating the stratigraphical sequence exist for Buca della Iena. When Stiner refers to “level E (called I5 in this study) of Buca della Iena [yielding] a U/Th date of approximately 40,000–41,000 BP, indicating that most of the assemblages (except level F, or I6 by my designation) are somewhat younger,” it is unclear which stratigraphical unit she is referencing. Based on available sources, it is evident that her level E refers to the flowstone. However, this unit was designated as level D in Fornaciari (1966) and level C in Pitti and Tozzi (1971), the only two published sources for Buca della Iena. Similarly, her level F should correspond to level D in Pitti and Tozzi and to level E in Fornaciari. Despite these inconsistencies, Stiner’s subdivision in six levels (I1–I6) appears to align with the stratigraphical sequence proposed by Pitti and Tozzi (A, B1, B2, B3, C, and D), excluding level E, which represents the basal rock layer. This correspondence is further suggested by the lithic counts she reports:

• Levels I1–I2 correspond to Pitti and Tozzi’s units A and B1,

• I3 to B2,

• I4 to B3,

• I5 to C,

• I6 to D

However, as we demonstrated in our manuscript, this stratigraphical division cannot be reliably followed, as bones and artefacts were marked only by sector and excavation spit from 1966, and the subsequent stratigraphical reassignment applied in Pitti and Tozzi (1971) was not applied in the archived collections. Overall, we believe we have provided sufficient evidence to demonstrate that Stiner’s analysis is not meaningfully comparable to ours due to the inconsistencies in stratigraphical unit designations. To clarify this issue, we have incorporated additional details on Stiner’s work in the Materials and Methods section of our manuscript.

Besides the total number of specimens involved also the size of the two samples should be mentioned, especially because sample sizes are very different.

We took care of implementing the reviewer's suggestion.

Info about methodologies used for assessing age at death and age categories should be added.

We took care of implementing the reviewer's suggestion.

Lines 165-166 cut marks are one type of butchery marks, so maybe just use the latter (similar occurrences are present also in other parts of the paper, so check them too).

We took care of implementing the reviewer's suggestion.

Line 168 it is “… gnawing by rodents and small carnivores…” and not “…of rodents…”

We took care of implementing the reviewer's suggestion.

Table 2 and Table 3. In both tables there are several lines with numbers but no categories. Maybe I am missing something, but in any case, the meaning of such numbers is not clear. Totals of the categories (e.g., Blade, Core) should be at least plural (or explicitly “Total Blades, Total Cores) and maybe placed at the bottom and not at the top of the corresponding list. Check numbers in the tables, for example in Table 2 the stratigraphically unreliable “Blade” (total) should be 8 and not 7, and actually the grand total in the last column is correctly 11 and not 10. For better clarity please add blade/flake/core/etc. wherever only the adjective is used.

We apologise for any confusion and have provided clearer tables and captions. Specifically, we have moved the totals below the subcategories for better readability. Additionally, we have addressed the issue of unnamed rows by implementing a structured text format:

Bold is used for broad technological categories (e.g., cores).

Italics indicate subcategories (e.g., Levallois cores).

Plain text is used for internal divisions of management products (e.g., Core edge flakes), which may be assigned to the Discoid or Levallois methods or remain unclassified.

Figures 6-8 and 12-14, for an easier reading, maybe add to the captions the reference to artifact categories at least by group, not leaving all the details to the supplementary materials.

The fact that at both sites “Complete artefacts are the most frequent” is related to “modern” selection during excavation and previous studies, or could it be ancient? Please discuss and comment on this. Is debris completely missing or was it not considered in this study.

We have modified the figures and the caption in accordance with the reviewer’s suggestion. Addressing questions about sample selection during excavation is somewhat challenging. We note that even undetermined faunal and lithic fragments (e.g., fragments) were collected, suggesting that no major purposeful selection was applied during excavation. However, the coarse excavation methods certainly did not allow for the collection of micro debris. The most parsimonious hypothesis is that site occupation intensity was relatively low, resulting in fewer artefacts being introduced compared to long-term palimpsests. This would have led to lower levels of artefact knapping activity, thereby resulting in a higher representation of complete artefacts than at most long-term palimpsest sites. Additionally, lower levels of trampling may have contributed to better preservation of the artefacts.

Lines 235-236. The reference to positive and negative peaks is not correct since there are no values below 0 in the graph of figure 11. Therefore, it is better to use “high” and “low”. (same in line 280). In order to assess artifact density how was the cubic meter calculated?

We have modified the definitions of peaks in accordance with the reviewer’s suggestion and apologise for any confusion caused.

Regarding the calculation of cubic metres, we did not initially provide an explanation as it follows a standard volumetric formula:

Volume=Length×Width×Height

To estimate artifact density, we multiplied the approximate area of the considered excavation sector by the reported height of the stratigraphical level under analysis. We acknowledge that this is a coarse measure, as finer details about the excavation remain uncertain. However, our conclusions regarding lithic densities aim to capture approximate variations throughout the sequence, which appear to correlate with the presence of anthropogenic butchery marks.

Furthermore, in calculating lithic densities, we followed established methodologies presented in published works that focus on this aspect (Clark and Barton, 2017; Bicho and Cascalheira, 2020). Notably, Bicho and Cascalheira (2020) refer to excavated volume as “Sampled Volume – volume of excavated sediments from the sampled area, presented in m³,” without detailing the specific methodology used to obtain this variable. Likewise, no further methodological details are provided in the references concerning the sites included (e.g., Lapa do Picareiro). Given this, we believe the most parsimonious interpretation is that their volumetric calculations followed the same approach as ours.

Bicho, N., & Cascalheira, J. (2020). Use of Lithic Assemblages for the Definition of Short-Term Occupations in Hunter-Gatherer Prehistory. In J. Cascalheira & A. Picin (Eds.), Short-Term Occupations in Paleolithic Archaeology (pp. 19–38). Cham: Springer International Publishing. https://doi.org/10.1007/978-3-030-27403-0_2

Clark, G.A., & Barton, C.M. (2017). Lithics, landscapes & la Longue-durée – Curation & expediency as expressions of forager mobility. Quaternary International, 450, 137–149. https://doi.org/10.1016/j.quaint.2016.08.002

We provide now a methodological explanation in the material and methods section.

In the captions of Figs 9-10 and 15-16. Better phrase “The frequency is reported as percentage using only stratigraphically reliable artefacts”.

For both sites it would be better to add a summary table with the total number of artifacts by level in order to understand better the statistical significance of the percentages reported in the graphs.

We have added a new SI file (SI file 10) showing a joint distribution of lithic artefacts and faunal remains by level.

In figures 11 and 17 the term levels instead of spits that is employed in all the other lithic graphs is used, while the fauna graphs for Buca della Iena use Level ….please unify. Mention somewhere in the first part that no lithics were found in layer 15 (it only appears much later in the paper).

We added a mention about the absence of lithics in level 15 of Buca della Iena. We also modified Figures 9, 10, 15, 16 so now we unified the terms.

Line 265 Assessing the homogeneity of an assemblage by comparing two samples of very different sizes (99 vs. 440) maybe is not so statistically valid.

Comparison and the conclusion of homogeneity are achieved through techno-typological assessment, not by sizes. As it is shown by the graphs and the text the same technological categories are recurring within both the stratigraphically reliable and unreliable contexts. Hence, we are reasonably confident that even if most artefacts are not in situ anymore, they most likely belong to the same period identified by the radiocarbon dates (i.e. the assemblages are Mousterian and do not belong to other technocomplexes).

Line 266 Table 5 is probably Table 4.

We apologise, we meant Table 3.

Line 321 Please add some comments and possibly a graph with the distribution of the faunal remains by layer.

Thank you for your suggestion. However, we believe that a graphical representation would be redundant, as the distribution of faunal remains by layer is already clearly specified in Table 4. We added a comment in the text about the distribution of faunal remains.

In the Fauna text and tables please correct Rhinoceratidae into Rhinocerotidae

We apologise and we corrected it according to the reviewer’s suggestion.

Line 332 Please specify in the methods which animals (at these sites) fall in the three size categories.

We provided the requested specifications in the Material and Methods section.

---

## [Decision Letter · Decision Letter 1]

2 May 2025

Buca della Iena and Grotta del Capriolo: new chronological, lithic, and faunal analyses of two late Mousterian sites in Central Italy

PONE-D-24-55044R1

Dear Dr. Gennai,

We’re pleased to inform you that your manuscript has been judged scientifically suitable for publication and will be formally accepted for publication once it meets all outstanding technical requirements.

Kind regards,

Enza Elena Spinapolice, Ph.D

Academic Editor

PLOS ONE

Additional Editor Comments (optional):

Reviewers' comments:

Reviewer's Responses to Questions

**Comments to the Author**

1. If the authors have adequately addressed your comments raised in a previous round of review and you feel that this manuscript is now acceptable for publication, you may indicate that here to bypass the “Comments to the Author” section, enter your conflict of interest statement in the “Confidential to Editor” section, and submit your "Accept" recommendation.

Reviewer #1: All comments have been addressed

Reviewer #2: All comments have been addressed

2. Is the manuscript technically sound, and do the data support the conclusions?

Reviewer #1: (No Response)

Reviewer #2: Yes

3. Has the statistical analysis been performed appropriately and rigorously? 

Reviewer #1: (No Response)

Reviewer #2: (No Response)

4. Have the authors made all data underlying the findings in their manuscript fully available?

Reviewer #1: (No Response)

Reviewer #2: Yes

5. Is the manuscript presented in an intelligible fashion and written in standard English?

Reviewer #1: (No Response)

Reviewer #2: Yes

6. Review Comments to the Author

Reviewer #1: (No Response)

Reviewer #2: (No Response)

7. PLOS authors have the option to publish the peer review history of their article (what does this mean? ). If published, this will include your full peer review and any attached files.

**Do you want your identity to be public for this peer review?** For information about this choice, including consent withdrawal, please see our Privacy Policy .

Reviewer #1: No

Reviewer #2: No

---

## [Editor Report · Acceptance letter]

PONE-D-24-55044R1

PLOS ONE

Dear Dr. Gennai,

I'm pleased to inform you that your manuscript has been deemed suitable for publication in PLOS ONE. Congratulations! Your manuscript is now being handed over to our production team.

Kind regards,

on behalf of

Dr. Enza Elena Spinapolice

Academic Editor

PLOS ONE